🔓 | **Open Peer Review** | Environmental Microbiology | Research Article

# Community- and genome-based evidence for a shaping influence of redox potential on bacterial protein evolution

**Jeffrey M. Dick,[1] Delong Meng[2]**

**ABSTRACT**    Despite deep interest in how environments shape microbial communities, whether redox conditions influence the sequence composition of genomes is not well known. We predicted that the carbon oxidation state ($Z_C$) of protein sequences would be positively correlated with redox potential (Eh). To test this prediction, we used taxonomic classifications for 68 publicly available 16S rRNA gene sequence data sets to estimate the abundances of archaeal and bacterial genomes in river & seawater, lake & pond, geothermal, hyperalkaline, groundwater, sediment, and soil environments. Locally, $Z_C$ of community reference proteomes (i.e., all the protein sequences in each genome, weighted by taxonomic abundances but not by protein abundances) is positively correlated with Eh corrected to pH 7 (Eh7) for the majority of data sets for bacterial communities in each type of environment, and global-scale correlations are positive for bacterial communities in all environments. In contrast, archaeal communities show approximately equal frequencies of positive and negative correlations in individual data sets, and a positive pan-environmental correlation for archaea only emerges after limiting the analysis to samples with reported oxygen concentrations. These results provide empirical evidence that geochemistry modulates genome evolution and may have distinct effects on bacteria and archaea.

**IMPORTANCE**    The identification of environmental factors that influence the elemental composition of proteins has implications for understanding microbial evolution and biogeography. Millions of years of genome evolution may provide a route for protein sequences to attain incomplete equilibrium with their chemical environment. We developed new tests of this chemical adaptation hypothesis by analyzing trends of the carbon oxidation state of community reference proteomes for microbial communities in local- and global-scale redox gradients. The results provide evidence for widespread environmental shaping of the elemental composition of protein sequences at the community level and establish a rationale for using thermodynamic models as a window into geochemical effects on microbial community assembly and evolution.

**KEYWORDS**    protein evolution, oxidation state, redox potential, Eh–pH diagram, global analysis, geochemistry, thermodynamics

Elemental stoichiometry of biomolecules affects elemental fluxes and trophic interactions in microbial ecosystems. However, relatively little is known about how environments may shape the elemental composition of protein sequences on evolutionary timescales. A deeper understanding of the relationship between environmental conditions and protein sequence evolution would complement efforts to develop geochemical proxies and molecular records of past environments (1, 2) and could provide new insight into the chemical factors that influence selection for taxa in microbial communities.

Address correspondence to Jeffrey M. Dick, jeff@chnosz.net, or Delong Meng, delong.meng@csu.edu.cn.

The authors declare no conflict of interest.

See the funding table on p. 20.

Evolutionary processes have had many millions of years to potentially be shaped by geological conditions. For example, within various genomes of opisthokonts (a major group of eukaryotes), the carbon oxidation states ($Z_C$) of proteins with gene ages assigned by phylostratigraphy—a technique in which conservation levels of orthologous genes among species are used to estimate ages of gene families—increase after the origin of cellular organisms at ca. 4.29 Ga (billion years ago), but the trends become more diverse as the lineages diverge after 1.1 Ga (3). The signal of oxidation derived from the elemental composition of protein sequences is consistent with the rise of atmospheric oxygen on Earth. A linkage between genome evolution and environmental conditions is conveyed by the hypothesis that, over evolutionary time, the elemental composition of genomically coded protein sequences may approach incomplete metastable equilibrium (a type of energy minimization) for given environmental conditions. Data for microbial communities enable new tests of this chemical adaptation hypothesis.

Genomic analyses have revealed intriguing patterns of elemental usage in protein sequences from different species (see reference [4] and references therein). However, observations for communities rather than species can be compared more directly with environmental measurements. Therefore, in this study, we analyzed community reference proteomes, which take into account the sequences of protein-coding genes from genomes available in the National Center for Biotechnology Information (NCBI) Reference Sequence (RefSeq) database, as well as the abundances of community members inferred from high-throughput 16S rRNA gene sequences. Actual protein abundances are not available in RefSeq, so reference proteomes for species obtained by summing the amino acid compositions of all coding sequences in the genome are based on the assumption that each sequence is equally important. Reference proteomes for species can be aggregated to higher taxonomic levels and then multiplied by taxonomic abundances to obtain community reference proteomes.

Although metatranscriptomic and/or metaproteomic data would be desirable to obtain a more accurate picture of the biomolecular composition of living communities, we chose to analyze community reference proteomes for two main reasons. First, we aimed to test the hypothesis of chemical adaptation of proteins at evolutionary timescales. Reference proteomes, or potentially expressed proteins, depend only on genome sequences, but metaproteomes, or actually expressed proteins, depend on genome sequences as well as on cellular physiology. By using reference proteomes, the analysis is focused on differences that emerge over evolutionary rather than physiological timescales. Second, data sets with redox potential measurements and 16S rRNA gene sequences are much more widely available than those for metagenomes, metatranscriptomes, or metaproteomes. To illustrate this semi-quantitatively, searches performed on Google Scholar on 26 December 2022 with the queries "16S rRNA" "redox potential", "metagenome" "redox potential", "metatranscriptome" "redox potential", and "metaproteome" "redox potential" gave about 15,200, 2,940, 521, and 212 results, respectively. Published 16S rRNA-based studies, therefore, provide the most extensive source of data that can be used to compare microbial communities with environmental measurements. Recently, comparisons of this type were performed for data sets for hydrothermal systems, stratified water bodies, and shale gas wells (5). That study identified multiple field sites where carbon oxidation state of community reference proteomes was aligned with measured oxygen concentrations; moreover, the 16S rRNA-based estimates show good correspondence with $Z_C$ of proteins inferred from shotgun metagenomes (5). However, to date, there has been no quantitative comparison of the carbon oxidation states of protein sequences with redox potential measurements at a global scale.

Concentrations of dissolved oxygen ($O_2$) diminish to levels that are below detection in many sediments (e.g., within several millimeters to centimeters of the sediment–water interface (SWI); see reference [6]) and other reducing environments. In anoxic settings, measurements of hydrogen ($H_2$) concentration can be used to monitor redox conditions (7). Oxygen and hydrogen, and other relevant oxidants and reductants, do not attain equilibrium in most environments. Nevertheless, a single redox scale is needed for a

global comparison of microbial habitats. Furthermore, the redox property should be detectable in all environments; neither $O_2$ nor $H_2$ concentration satisfies this criterion, but electrode measurements do. For this reason, a measurement referred to as oxidation–reduction potential or redox potential (Eh) was selected in this study. Redox potential denotes the tendency for oxidation–reduction reactions to occur, with higher (more positive) values indicating a greater propensity for loss of electrons. The Eh scale is expressed as an electrical potential (i.e., voltage) with reference to the standard hydrogen electrode (SHE). Field measurements typically use a redox probe consisting of a platinum indicator electrode and an internal Ag/AgCl reference electrode connected to a potentiometer. Redox probes must be periodically calibrated with solutions of known redox potential (e.g., ZoBell's solution), and a temperature-dependent conversion is used to convert the meter readings from volts versus Ag/AgCl to volts versus SHE (8). A rigorous chemical interpretation of Eh measurements in natural systems is challenging because they represent mixed potentials that are affected by the presence of various, often not well-defined, electroactive species that are generally not in mutual equilibrium (9). Despite its chemical complexity, Eh appears in many environmental microbiology studies, calling for new interdisciplinary approaches to examine the ecological relevance of this parameter.

The specific aims of this study are first, to develop a multiscale (local to global) perspective on differences of carbon oxidation states of community reference proteomes in relation to redox potential; second, to characterize these differences in phylogenetic domains (bacteria and archaea); and finally to examine practical issues such as whether Eh or $O_2$ concentration is a better predictor of carbon oxidation state, the robustness of the results to different primer sets for the 16S rRNA gene, and comparison with metaproteomic data. The results provide a novel hypothesis-driven picture of global microbial communities as chemical entities and develop the rationale for integrating geochemical thermodynamics into evolutionary models for microbial ecosystems.

## MATERIALS AND METHODS

### Thermodynamic calculations

We used group additivity parameters that include pH-dependent ionization of side-chain and terminal groups (10) as implemented in the CHNOSZ package (11) to calculate standard Gibbs energies from amino acid compositions of reference proteomes. Chemical formulas of reference proteomes were normalized by the total number of amino acids, thus eliminating protein length as a variable in the calculations. The chemical system was defined using a minimum number of thermodynamic components, also known as basis species. We modified the previously described QEC basis species (glutamine, glutamic acid, cysteine, $H_2O$, and $O_2$) (12) by substituting $H^+$ and $e^-$ for $O_2$; the addition of electronic charge is needed for calculating an Eh–pH diagram. The chemical activities ($a$) of some of the basis species were assigned constant values (log$a$ Gln = −3.2, log$a$ Glu = −4.5, log$a$ Cys = −3.6, and log$a$ $H_2O$ = 0), leaving Eh and pH as variables for making a relative stability diagram. The relative stability fields on this diagram represent the reference proteome with the lowest Gibbs energy of formation from the basis species (i.e., the least unstable reference proteome) compared to the others.

### Data sources

We used literature searches to find field-based data for different environments and field- and laboratory-based data (including mesocosms) for soils and sediments; experiments described as bioleaching were not included. All published data sets that we could find as of July 2022 were included in our compilation if (i) there were sufficient metadata to match sample names to database accession numbers, (ii) demultiplexed 16S rRNA gene sequences were available in the NCBI Sequence Read Archive (SRA) (13), and (iii) there were at least 20 biological samples with an associated Eh range of at least

100 mV before correction to pH 7. The sample number cutoff was reduced to 15, 10, and 8 for groundwater, geothermal, and hyperalkaline environments, respectively, to compile enough data sets for global comparison. Data set names used here, NCBI BioProject accession numbers, and references are listed in Table 1. Each study (i.e., source publication) corresponds to one data set, with the following exceptions: two data sets were compiled from each of the studies of Ghannam et al. (14) (Port Microbes, the two data sets are for samples collected on a prefilter or postfilter), and Power et al. (15) (New Zealand Hot Springs, the two data sets are for acidic or circumneutral to alkaline samples). Data for Winogradsky columns (16, 17) were analyzed separately from the main compilation.

$O_2$ concentration and redox potential, the latter assumed to be stated relative to SHE, were taken from the primary publications (see Table S1 for details). Where needed, values were extracted from figures using g3data (https://github.com/pn2200/g3data, accessed on 11 October 2022). Temperature ($T$) and pH were also tabulated if available; otherwise their values were assumed to be 25°C and pH 7 for the conversion to Eh7 (see below). Where there are extreme values of temperature and pH, such as in hot springs and

**TABLE 1** Data sets analyzed in this study[a]

| River & seawater |
|---|
| Pearl River Estuary (PRJNA319446 and PRJNA357334 [18]), Sansha Yongle Blue Hole (PRJNA503500 [19, 20]), Port Microbes–prefilter and postfilter (PRJNA542685 and PRJNA542890 [14]), Bahe River (PRJNA588356 [21]), Nu River (PRJNA663210 [22]), Maozhou River (PRJNA681688 [23]), Plastisphere (PRJNA717904 [24]), Three Gorges Reservoir (PRJNA733826 [25]), and Taxco AMD (PRJNA801253 [26]) |

| Lake & pond |
|---|
| Microbial Mat, Kiritimati Atoll (PRJNA174394 [27]), Xidong Reservoir, Xiamen (PRJNA315049 [28]), Ursu Lake, Romania (PRJNA395513 [29]), Xiamen Reservoirs and Ponds (PRJNA407260 [30]), Keweenaw Waterway, Michigan (PRJNA489447 [31]), German Kettle Holes (PRJNA641761 [32]), Lake Kinneret, Israel (PRJEB39923 [33]), Monegros Desert, Spain (PRJNA429605 [34]), and Lake Varese, Italy (PRJNA694444 [35]) |

| Geothermal areas |
|---|
| New Zealand Hot Springs–acidic and circumneutral to alkaline (PRJEB24353 [15]), Eastern Tibetan Plateau (PRJNA592622 [36]), Uzon Caldera (PRJNA623081 [37]), and Southern Tibetan Plateau (PRJNA638734 [38]) |

| Hyperalkaline fluids |
|---|
| CROMO 1 (PRJNA289273 [39]), Samail Ophiolite (PRJNA352492 [40]), Santa Elena Ophiolite (PRJNA361138 [41]), Voltri Massif (PRJNA685937 [42]), CROMO 2 (PRJNA690585 [43]), and Samail Ophiolite Packers (PRJNA743134 [44]) |

| Groundwater |
|---|
| Sarnia nZVI Injection (PRJNA308958 [45]), Hetao Basin (PRJNA350383 [46]), Jianghan Plain (PRJNA377933 [47]), Ohio Aquifers (PRJNA387583 [48]), Rayong Province (PRJNA434769 [49]), Mezquital Valley (PRJNA488796 [50]), Hainich Critical Zone (PRJEB33032 [51]), Po Plain (PRJNA667833 [52]), New Zealand Aquifers (PRJNA699054 [53]), and Aquifer SE of Melbourne (PRJNA861729 [54]) |

| Sediment |
|---|
| Mai Po Wetland (PRJEB12429 and PRJEB12432 [55]), **Baltic Sea Swedish Coast** (PRJNA322450 [56]), **Finnish Boreal Lakes** (PRJNA349972 [57]), Honghu Lake (PRJNA352457 [58]), Marine and Freshwater Sediments (PRJNA393823 [59]), Daya Bay (PRJNA400089 [60]), Lake Hazen, Nunavut (PRJNA430127 [61]), Jinchuan River (PRJNA437688, PRJNA437695, PRJNA437692, and PRJNA437697 [62]), Lake Neusiedl (PRJNA507590 [63]), **Hydrocarbon Biodegradation** (PRJNA523725 [64]), Bay of Biscay (PRJEB35647 [65]), Aldabra Atoll (PRJNA611521 [66]), Maozhou River Hyporheic Zone (PRJNA616197 [67]), **Estuarine Sediment Mesocosms** (PRJNA639965 [68]), **Hydrodynamic Experiments** (PRJNA777293 [69]), and Cardinal Pond, Poland (PRJNA832534 [70]) |

| Soil |
|---|
| Dabu Town, Hunan Province (PRJNA361046 [71]), **Urban Wetland Soils** (PRJNA415514 [72]), Tengger Desert Biocrusts (PRJNA543295, PRJNA647192, and PRJNA647699 [73]), **Paddy Soil Water Management** (PRJNA564714 [74]), **Anaerobic Soil Disinfestation** (PRJNA575041 [75]), East Asia Paddy Soil (PRJNA607877 [76]), **Rice Rhizosphere** (PRJNA611687 [77]), Georgia Salt Marshes (PRJNA666636 [78]), **Paddy Soil Amendments 1** (PRJNA690162 [79, 80]), Maize Intercropping, Poland (PRJNA725644 [81]), **Paddy Soil Amendments 2** (PRJDB12684 [82]), and **Soil–Water Interface** (PRJNA826420 [83]). |

| Metaproteome comparisons |
|---|
| Manus Basin Inactive Chimney (PRJEB27164 [84]), Manus Basin Active Chimneys (PRJEB5213 [85]), Soda Lake Biomats (PRJNA377096 [86]), Mock Communities (PRJEB19901 [86]), and Saanich Inlet (PRJNA247822 [87]). |

[a] The name of each data set is followed by the BioProject accession number(s) and one or two literature references; the second reference, if present, is for redox potential and/or $O_2$ data. Bold text is used for laboratory or mesocosm experiments. Abbreviations: AMD, acid mine drainage; CROMO, Coast Range Ophiolite Microbial Observatory; nZVI, nanoscale zero-valent iron.

hyperalkaline systems, these variables are routinely reported; for the few data sets for non-extreme environments where neither $T$ nor pH was reported (river & seawater: 1, lake & pond: 1, sediment: 4, soil: 2; see Table S1), their effects on Eh7 are likely to be relatively small. Brief descriptions of samples (e.g., substrate type, sample location, or collection time) were also recorded as applicable to each data set.

## Correction of Eh to pH 7

Because of the dependence of electrochemical potential on pH, known as the Nernst slope or Nernst factor, reported Eh values were corrected to pH 7 with (88)

$$\text{Eh7} = \text{Eh} + \frac{d\text{Eh}}{d\text{pH}} \times (7 - \text{pH}) \tag{1}$$

where Eh7 denotes the corrected value and $d$Eh/$d$pH is the theoretical slope, which is equal to −59.16 mV/pH unit at 25°C. Values of the Nernst slope at other temperatures were calculated from −2.303 $RT/F$, where $R$, $T$, and $F$ are the gas constant, temperature in Kelvin, and Faraday constant. The theoretical relationship between pH and Eh as expressed in Equation 1 may not hold for all natural systems (89), so correcting Eh to pH 7 is not recommended for reporting primary results (90).

## Unit conversion of oxygen concentrations

$O_2$ concentrations reported in units of milligram per liter were converted to micromoles per liter using the molar mass of $O_2$ (31.9988 g/mol). Concentrations reported as percent saturation (%) were converted to micromoles per liter by first combining Henry's constant for $O_2$ at the reported sample temperature, or 25°C if not available, with the partial pressure of $O_2$ in the atmosphere at sea level (0.21 atm = 0.2128 bar) to calculate saturated $O_2$ concentration, which was then multiplied by the percentage value to obtain concentration in micromoles per liter. Henry's constants as a function of temperature were calculated using the subcrt function in the CHNOSZ R package version 1.4.3 (11).

## Sequence data processing

One data set in our compilation is for environmental 16S amplicon rRNA (i.e., not shotgun) sequences reverse-transcribed to cDNA (Soil–Water Interface [83]); it was processed in the same way as the other data sets, but taxonomic abundances were derived from the taxonomic classifications of cDNA sequences. All other data sets are for 16S rRNA gene sequencing of environmental DNA; if results for cDNA were also reported, they were not included in our compilation. Three large data sets were subsampled so that they would not dominate the global analysis. The complete New Zealand Hot Springs data set consists of 925 samples (15); we used data for 42 randomly selected acidic samples (pH < 3) and 39 randomly selected circumneutral to alkaline samples (pH > 6). For the Bay of Biscay data set (65), we used data for the 47 sediment samples with redox potential measurements collected in 2017, which is a subset of the 256 samples collected in different years. For the Port Microbes data set (14), we used data for the first three prefilter and first three postfilter samples with at least 20,000 sequenced read pairs and complete metadata for each port (114 of 1,373 available samples).

Sequence data were processed with a custom pipeline consisting of merging of forward and reverse reads where applicable, quality and length filtering, singleton removal, reference-based chimera detection, and taxonomic classification. The R script implementing the pipeline is available (see Data availability). The pipeline uses the following software and databases: fastq-dump from the NCBI SRA Toolkit for generating FASTQ files from downloaded SRA files, VSEARCH (91) for merging, singleton and chimera detection, seqtk (https://github.com/lh3/seqtk, accessed on 26 April 2023) for extracting non-singletons, SILVA SSURef NR99 database version 138.1 (92) for chimera detection, and RDP Classifier version 2.13 (93) for taxonomic classification with the included training

set (RDP 16S rRNA training set no. 18 07/2020) and the default confidence threshold of 80%.

For the two 454 pyrosequencing data sets in our compilation (27, 45), length filtering was specified as a minimum read length of 300 and maximum length of 500 (-fastq_minlen and -fastq_maxlen options of VSEARCH); this setting is based on the processing steps described by Kocur et al. (45). To process sequence data from the Illumina and Ion Torrent platforms, we adjusted the truncation length (-fastq_trunclen option of VSEARCH) to maximize the performance of the taxonomic classifier. Specifically, a truncation length was individually selected for each data set as the highest number, rounded to the nearest 10, where the majority of reads passed the filtering step. After length and quality filtering and singleton removal, runs were subsampled to a depth of 10,000 sequences before chimera detection and taxonomic classification; the subsampling was done only to reduce processing time and is not the same as normalization of sequencing depth used for computing diversity metrics (see Methodological limitations and justification). Settings for each data set, sequence processing statistics, and additional details are listed in Table S2.

## Taxonomic mapping and amino acid composition of community reference proteomes

RDP classifications at the root or domain level or to chloroplast or eukaryota were omitted. No filtering of mitochondrial reads was done because no sequences labeled as such are present in the training set included with the RDP Classifier. The lowest assigned taxonomic level (from genus to phylum) for each remaining sequence was mapped to the NCBI taxonomy represented in RefSeq by automatic matching of both rank and taxon names or by manual mapping for particular taxa as described previously (5). The following within-level manual mappings were used (RDP → NCBI): genus, *Escherichia/Shigella* → *Escherichia*, Gp1 → *Acidobacterium*, Gp6 → *Luteitalea*, GpI → *Nostoc*, GpIIa → *Synechococcus*, GpVI → *Pseudanabaena*; family, Family II → *Synechococcaceae*, *Ruminococcaceae* → *Oscillospiraceae*; order, *Clostridiales* → *Eubacteriales*, *Rhizobiales* → *Hyphomicrobiales*; class, *Planctomycetacia* → *Planctomycetia*; phylum, *Cyanobacteria/Chloroplast* → *Cyanobacteria*. Several cross-level manual mappings were also used: genus Subdivision3_genera_incertae_sedis → family *Verrucomicrobia* subdivision 3, genus Spartobacteria_genera_incertae_sedis → class *Spartobacteria*, class *Cyanobacteria* → phylum *Cyanobacteria*, and class *Actinobacteria* → phylum *Actinobacteria*. No manual mapping was done for archaeal taxa. Percentages of mapped taxa are listed in Table S2. Unmapped taxonomic assignments were omitted from subsequent analysis.

For sequencing runs for libraries generated using domain-specific bacterial or archaeal primers, only the taxonomic assignments in the respective domain were kept. For studies that used universal primers, the RDP Classifier assignments within each domain were selected. Samples with less than 100 sequences with lowest-level taxonomic assignments from genus to phylum level for either bacteria or archaea were excluded from downstream analysis. If the archaeal metacommunity in any data set was represented by fewer than four remaining samples, it was not processed further.

We used previously compiled reference proteomes (5) for archaeal and bacterial taxa at levels from genus to phylum derived from the NCBI RefSeq database release 206 (94). To generate the reference proteomes, only species-level NCBI taxonomic IDs (taxids) were used, and archaeal and bacterial species with less than 500 reference protein sequences were excluded. For each species-level taxid, the sum of amino acid compositions of all reference sequences was divided by the number of reference sequences to obtain the mean amino acid composition; this was done so that species with different proteome sizes contribute equally to the reference proteomes of higher-level taxa. For each genus, the mean amino acid compositions of all species-level taxa in that genus were summed and divided by the number of species to obtain the amino acid composition of the reference proteome. Analogously, the mean amino acid compositions of all

species in each family, order, class, and phylum were averaged to obtain the reference proteomes for taxa at those levels.

Within each domain, the counts of all mapped lowest-level taxonomic assignments were multiplied by the amino acid compositions of the corresponding reference proteomes and divided by the total count of mapped assignments to obtain the amino acid composition of the community reference proteome.

## Metaproteomes

Publicly available processed metaproteomic data were used. Except as noted below, protein IDs in pep.xml or mztab files, excluding sequences labeled as contaminants and reverse decoy sequences, were matched to the protein reference database for that study to get protein sequences. For pep.xml files, the first Protein ID for each mass spectrum search query was used; protein abundance was estimated by counting multiple identifications of the same protein in different peptide spectra. For mztab files, the amino acid composition of each identified protein was multiplied by the number of peptide spectral matches. The amino acid compositions of all proteins were then summed and used to calculate $Z_C$. The ProteomeXchange accession numbers and specific files used are listed next. *Manus Basin Inactive Chimney* (84): PXD010074; protein sequence database: DEAD_Chimneys_20170720_nr_fw_rev_cont.fasta; protein IDs: Mudpit_121204_V2_P6_CH_SM_Chimney_3b_1.pride.mztab.gz. *Manus Basin Active Chimneys* (95): PXD009105; protein sequence database: 140826_StM_Chimney39surface53surface_fw_rev_cont.fasta; protein IDs: Mudpit_121026_V2_P6_AO_SM_Chimney_1a_10.pride.mztab.gz (diffuse-flow chimney RMR-D) and Mudpit_121028_V2_P6_AO_SM_Chimney_2a_1.pride.mztab.gz (focused-flow chimney RMR5). *Soda Lake Biomats* (86): PXD006343; protein sequence database: SodaLakes_AllCombined_Cluster95ID_V2.fasta; protein IDs: GEM-(01).pep.xml and LCM-(01).pep.xml. *Mock Communities* (86): PXD006118; protein sequence database: Mock_Comm_RefDB_V3.fasta; protein IDs: 12 pep.xml files (three mock communities with four replicates each for 260-min 1D-LC-MS/MS runs). *Saanich Inlet* (87): PXD004433; protein sequence database: SaanichInlet_LP_ORFs_2015-01-13_Filtered.fasta; protein IDs: SBI_Metagenome2015_AllProteinsAllExperiments.txt (the ScanCount column in this table was used to estimate protein abundance). Metaproteomic experiment names were mapped to sample names using Table S2 of reference 87.

To obtain community reference proteomes for comparison with metaproteomes, 16S rRNA gene sequencing data sets were taken from the studies cited above, except for Manus Basin Active Chimneys (85).

## Computational methods

Average oxidation state of carbon ($Z_C$) for proteins can be calculated from (12)

$$Z_C = \frac{-h + 3n + 2o + 2s}{c} \qquad (2)$$

where the lower case letters *c*, *h*, n, *o*, and *s* are the numbers of the respective elements in the elemental formula. Rather than using Equation 2 directly, $Z_C$ was calculated by combining amino acid compositions of community reference proteomes with precomputed $Z_C$ values for amino acids; the calculation includes weighting by the number of carbon atoms in each amino acid (12).

Carbon oxidation state was computed from amino acid composition using the ZCAA function in the canprot R package version 1.1.2 (https://cran.r-project.org/package=canprot), and linear regressions were performed using lm in R version 4.2.1 (96). The world map, drawn with the Winkel Tripel projection, was made using the oce R package version 1.7-2 (97) and its included coastlineWorld data set together with shapefiles for the North American Great Lakes (98).

## Statistics

We subjected one hypothesis to multiple trials by assessing correlations between Eh7 and $Z_C$ obtained from different local and global data sets. Alternative hypotheses using Eh or $O_2$ concentration instead of Eh7 were only tested for the pan-environmental global data set. Except for the subsampling of samples and sequencing reads described above, no data were removed from the analysis; in particular, outliers were not removed. The statistical analyses used are the slopes of linear regressions (*m*) and Pearson correlation coefficients (*r*). Values of the slope are reported with margin of error (*MOE*) as $m \pm MOE_{95}$, where $[m - MOE_{95}, m + MOE_{95}]$ is the 95% CI; this was used in lieu of *P*-values for correlations. To assess the statistical significance of the frequency of positive results, *P*-values were calculated using exact one-sided binomial tests assuming a 50% chance of a positive correlation.

## Methodological limitations and justification

The assignment of entire data sets to environment types may include samples that do not all fit that description. For example, some sediment data sets also contain samples of overlying water (see Table S1). A more fine-grained approach would be to describe environment types at the level of samples rather than data sets. However, for the purposes of local-scale comparisons, including all samples in each data set is more faithful to the source study designs, some of which include different sample types that are used as controls. We feel this advantage outweighs the negative impact on the global-scale comparison, especially since the number of potentially misclassified samples in the global analysis is relatively small.

We did not use operational taxonomic units (OTUs) or amplicon sequence variants (ASVs) in our analysis. A common reason for using OTUs or ASVs is to generate diversity metrics, including various metrics for alpha and beta diversity. In contrast, the calculation of chemical metrics only requires estimates of total elemental composition at the community level. A straightforward approach is to add up the elemental compositions of the reference proteomes corresponding to each taxonomically classified sequence without clustering. Because the most abundant taxa dominate the elemental composition, variability of sequencing depth among data sets is not a major source of uncertainty in the calculation of chemical metrics.

We used the default RDP Classifier training set for taxonomic classification rather than the more popular SILVA database. The SILVA database is larger and might provide greater taxonomic resolution (i.e., a higher proportion of genus-level assignments). Lower taxonomic resolution tends to lessen the differences of $Z_C$ (see Fig. S2 in reference 5); therefore, this limitation is more likely to generate false-negative than false-positive results.

There are limitations inherent in the use of reference proteomes and taxonomic mapping; reference proteomes contain no information about protein expression levels, and mapping between the RDP and NCBI taxonomies is somewhat uncertain. To overcome the limitations of a taxonomy-based approach, future work should consider whether phylogenetic inference methods, which are currently available for inferring functional traits (99), could be modified to infer amino acid composition. However, phylogenetic inference requires a reference database of traits, so limited knowledge of protein expression levels would still be an issue.

## RESULTS

### Theoretical background: incomplete metastable equilibrium

The theoretical relation between environmental variables and molecular oxidation state can be quantified by using methods adapted from geochemical thermodynamics. Under ambient conditions, proteins are thermodynamically unstable relative to smaller molecules and tend to spontaneously (i.e., exergonically) hydrolyze into amino acid monomers, which themselves are unstable relative to inorganic compounds. By

excluding from consideration the potential for the formation of amino acids and other small molecules, a metastable equilibrium model for reference proteomes can be constructed. The model described here is defined in terms of thermodynamic components (see reference [100]), which are mathematical abstractions that theoretically relate the elemental composition of reference proteomes to chemical variables such as Eh and pH but do not necessarily reflect actual mechanisms of protein synthesis or evolution.

For the purposes of illustration, we selected reference proteomes for five methanogen species (*Methanococcus vannielii*, *Methanococcus maripaludis*, *Methanococcus voltae*, *Methanobrevibacter smithii*, and *Methanofollis liminatans*) having a range of $Z_C$ from −0.220 for *M. vannielii* to −0.154 for *M. liminatans* (see Fig. 1 in reference 101). These species represent a diverse phylogeny, including both Class I and Class II methanogens, but they are all mesophiles with optimal growth temperatures of 30°C–40°C. We wrote overall formation reactions for the reference proteomes from thermodynamic components, calculated the Gibbs energies of the reactions as a function of pH and Eh at 25°C, then identified the reference proteome with the lowest Gibbs energy to plot relative stability fields (Fig. 1a; for further details, see Materials and Methods: Thermodynamic calculations). Because the Gibbs energies are positive except at the most reducing conditions (near the stability limit of water), the reference proteomes are unstable relative to the thermodynamic components across most of the diagram, and "relative stability" denotes the least unstable proteome compared to the others.

The reference proteomes with the lowest and highest $Z_C$ are relatively stable at low and high Eh, respectively, and cross sections of the Eh–pH diagram at discrete values of pH exhibit a stepwise increase of $Z_C$ with increasing Eh (Fig. 1b). Because of the negative slopes of the stability-field boundaries on the Eh–pH diagram, lower pH is associated with higher Eh, and vice versa. To allow comparing observations at different pHs, a quantity known as Eh7 can be used (88); it is calculated by moving along a line of constant slope from a given pH to pH = 7 (equation 1). Values of $Z_C$ in the metastable equilibrium calculation are widely dispersed along the Eh scale at three pHs from 5 to 9 but overlap when plotted against Eh7 (Fig. 1b). It should be noted that the relative stabilities of proteomes depend on both pH and Eh; the correction to Eh7 is independent from the metastable equilibrium calculation and is only used to illustrate how values of Eh obtained at different pH values can be compared.

Real biological systems are not at complete metastable equilibrium. Chemical adaptation is simply the hypothesis of an incomplete equilibrium. We simulated non-equilibrium biological effects by assigning Eh7 values at equal intervals, then picking a random sample of species from the RefSeq database for which the slope of the regression between the assigned values of Eh7 and $Z_C$ of the species' reference proteomes is close to zero (Fig. 1c). Then, we calculated a weighted mean of $Z_C$ assuming weights of 20% for the metastable equilibrium values and 80% for those of the randomly sampled species. The addition of a random effect decreases the slope and widens the confidence interval of the slope. The slope calculated in this simulation (0.133/V) is comparable to the highest slopes empirically observed for local-scale data sets below. This simple simulation highlights the feasibility of detecting incomplete metastable equilibrium against a background of random biological variation.

To generalize from this model, thermodynamic considerations predict a positive association between $Z_C$ and Eh7. The aim of this study is to empirically test this prediction using data for natural communities, which can unveil not only positive but also negative or non-significant correlations. The analysis below uses data for modern communities and genomes, but our interpretation is about evolutionary differences, similar to other comparative genomic studies (102). Although the hypothesis of chemical adaptation presupposes a sufficient period of genome evolution under steady redox conditions, the prediction may also be applicable to short-term experimental interventions because of the phenomenon of species sorting—that is, environmental selection for preadapted species (103).

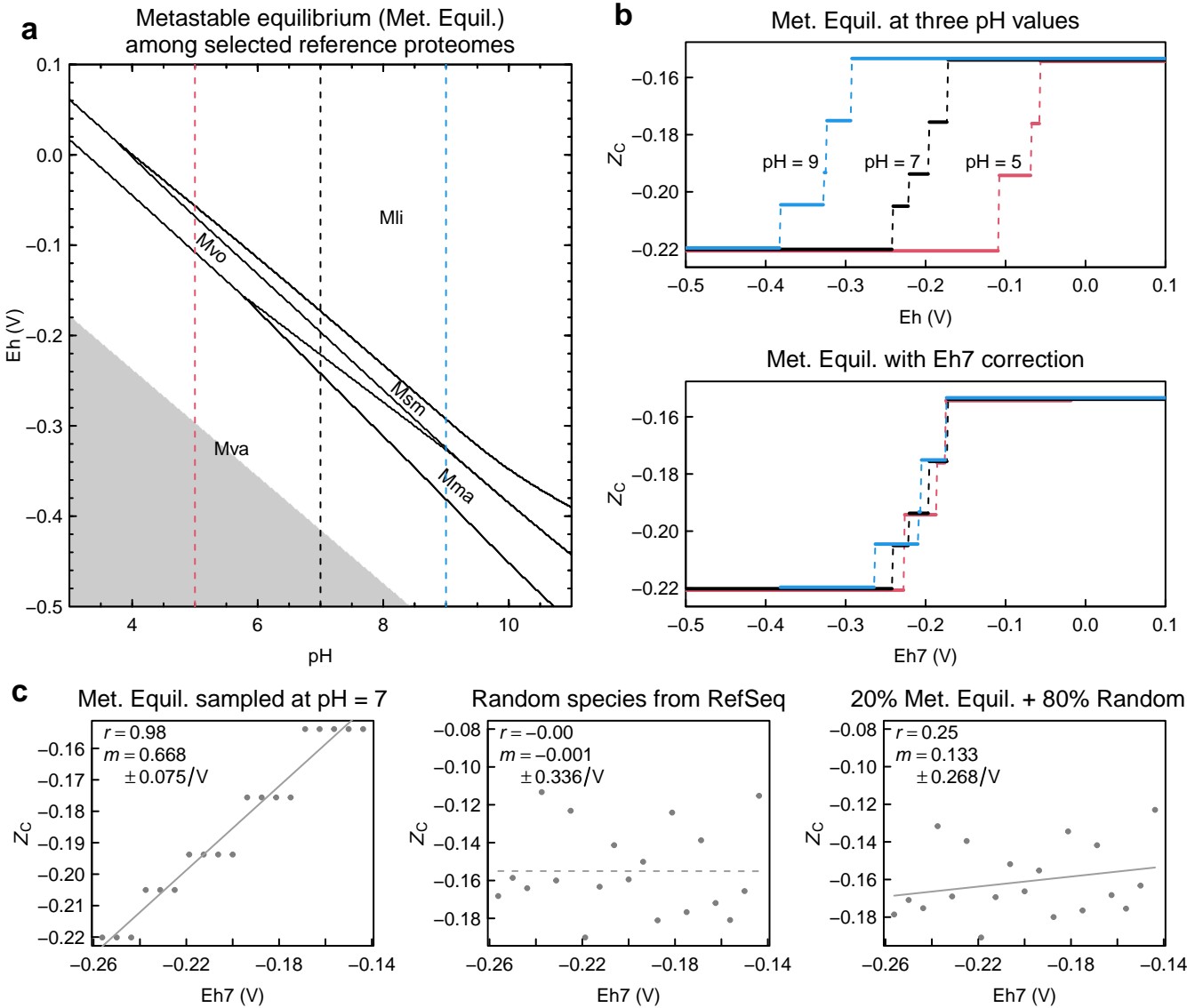

**FIG 1** Thermodynamic model for the relationship between carbon oxidation state of reference proteomes and redox potential. (a) Eh–pH diagram for reference proteomes of five selected methanogen species. All proteomes are unstable relative to the basis species used in the calculations (glutamine, glutamic acid, cysteine, $H_2O$, $H^+$, and $e^-$); the relative stability fields, therefore, represent the proteome that is least unstable at particular values of Eh and pH. The gray area represents reducing conditions that are beyond the stability limit of water. (b) Carbon oxidation state of relatively stable reference proteomes as a function of Eh or Eh7 at three pH values. The upper plot shows $Z_C$ of the most stable reference proteome at three pH values, corresponding to the vertical dashed lines in (a). Note, for instance, that the reference proteome for Mvo, which is relatively stable at pH = 5, has $Z_C$ that is intermediate between those of Mma and Msm, which are relatively stable at pH = 9. The lower plot shows the same profiles after correction to Eh7 using equation 1. (c) A simple conceptual model for equilibrium and non-equilibrium effects on the relation between carbon oxidation state and Eh7. The first plot shows theoretical equilibrium values of $Z_C$ sampled at equal intervals of Eh7, taken from the profile for pH = 7 in (b). The second plot shows $Z_C$ for randomly selected archaeal and bacterial species from RefSeq, representing non-equilibrium biological variability; this particular choice of random species displays essentially zero correlation in the plot. The third plot shows a linear combination of the first and second plots. Abbreviations: Mli, *Methanofollis liminatans*; Mma, *Methanococcus maripaludis*; Msm, *Methanobrevibacter smithii*; Mva, *Methanococcus vannielii*; Mvo, *Methanococcus voltae*.

## $Z_C$ of reference proteomes reflects oxygen tolerance and is distinct from metaproteomes

Various microorganisms have distinct oxygen tolerance. The "List of Prokaryotes According to Their Aerotolerant or Obligate Anaerobic Metabolism" (version 1.2) (104) classifies 644 prokaryotic genera as strictly anaerobic or aerotolerant, including

facultative anaerobes. This list is predominated by bacteria, with two archaeal genera present in RefSeq (*Methanobrevibacter* and *Methanosphaera*). We found that the carbon oxidation state of reference proteomes for aerotolerant genera is significantly higher compared to strict anaerobes (Fig. 2a). This finding is broadly consistent with the notion that genomes code for protein sequences that are to some extent chemically adapted to redox conditions.

Next we aimed to find out whether estimates of carbon oxidation state for community reference proteomes are similar to values calculated for actual metaproteomes. The data sets were selected with the criteria of joint availability of metaproteomic data (including a protein sequence database) and 16S rRNA gene sequences. The analyzed data sets include active and inactive chimneys at the Manus Basin hydrothermal vent field (84, 85), the water column of the Saanich Inlet in British Columbia, Canada (87), and soda lake biomats from the Rocky Mountains in Canada (87). We also analyzed data for mock communities generated in the latter study (87); although the species composition of the mock communities was known, we did not use that information but instead generated the community reference proteomes from 16S rRNA gene sequences.

The proteins identified in the metaproteomic studies were used to calculate $Z_C$ (see Materials and Methods: Metaproteomes). In parallel, 16S rRNA gene sequences for the same samples were used to generate community reference proteomes. Because of limitations in mapping Ribosomal Database Project (RDP) classifications to the NCBI taxonomy (see Materials and Methods: Taxonomic mapping and amino acid composition of community reference proteomes), a reference proteome for each identified taxon in a community was not always available. The percentages of mapped taxa are listed in Table S2. The community reference proteomes were calculated by multiplying the amino acid composition of the reference proteome for each identified taxon by the taxonomic abundance and omitting those taxa without available reference proteomes.

There is a weak positive correlation between metaproteomes and community reference proteomes, but a generally higher $Z_C$ of metaproteomes (Fig. 2b). In experiments designed to provide an unbiased estimate of membrane protein abundance,

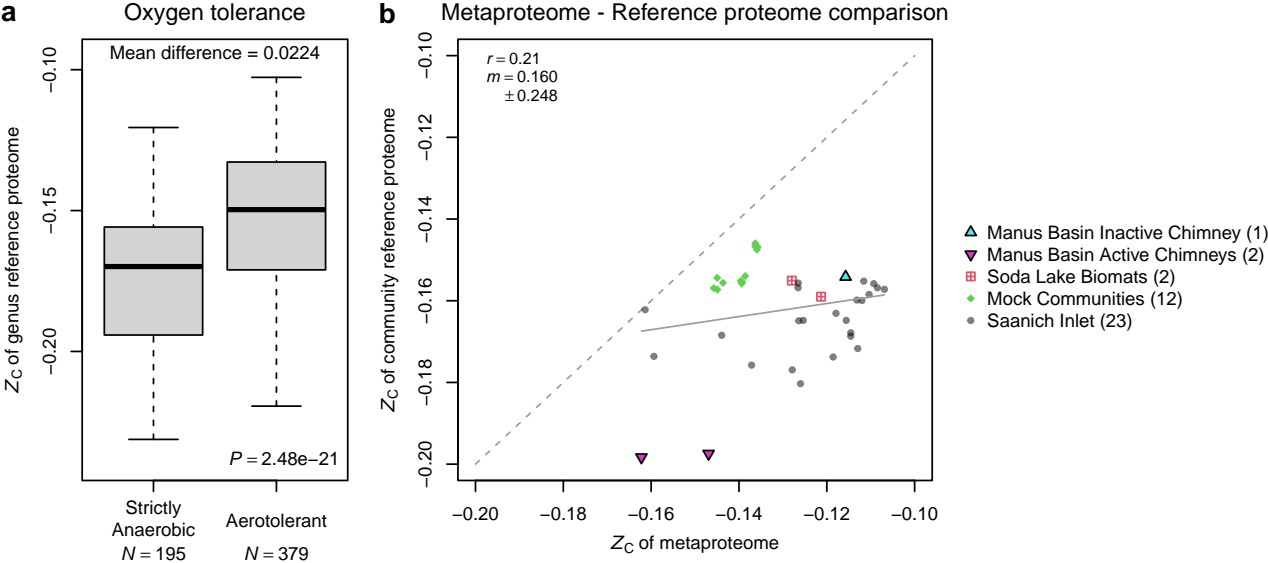

**FIG 2** $Z_C$ of reference proteomes compared with oxygen tolerance and with metaproteomes. (a) Carbon oxidation state ($Z_C$) of reference proteomes for strictly anaerobic and aerotolerant genera in the RefSeq database. The oxygen tolerance of prokaryotic genera was taken from Table S1 of reference (104). Of the genus names in that table, 64 were not matched to RefSeq and were omitted from the comparison. Student's two-sided *t*-test was used to calculate *P* value. (b) Comparison of metaproteomes and community reference proteomes. $Z_C$ was calculated for all proteins identified in each metaproteome. Independently, $Z_C$ was also calculated for community reference proteomes derived from 16S rRNA sequences and RefSeq proteomes. The dashed line is the theoretical 1:1 line; the solid line shows linear regression of all data points with Pearson correlation coefficient (*r*) and slope (*m*) indicated in the legend.

the total abundance of cytoplasmic proteins in *Escherichia coli* is more than double than that of membrane proteins (data from Table S3A of reference 105). Furthermore, common mass spectrometry (MS)–based proteomics workflows have limited ability to detect membrane proteins (106). Because they are enriched in hydrophobic amino acids, membrane proteins tend to be more reduced than cytoplasmic proteins (107). For instance, in the yeast *Saccharomyces cerevisiae*, sequences of proteins localized to the cytoplasm and plasma membrane have mean $Z_C$ of −0.127 and −0.188, respectively (107). Therefore, technical bias against detecting membrane proteins in MS-based proteomics and the higher natural abundance of cytoplasmic than membrane proteins are two factors that could possibly contribute to the higher $Z_C$ of proteins observed in metaproteomes compared to community reference proteomes.

An interesting result is that active hydrothermal chimneys have the most reduced community reference proteomes among the data sets compared in Fig. 2b. This result shows that the highly reducing environment associated with hydrothermal fluids corresponds to relatively reduced community reference proteomes. Notably, metaproteomes indicate that the expressed proteins of communities in active chimneys are also reduced relative to those in most other environments. Unfortunately, metaproteomic data were not available for assessing correlations with redox potential in the remainder of this study.

## Winogradsky columns

Based on the above considerations, we predicted to find a positive correlation between redox potential and carbon oxidation state of community reference proteomes for various environments (Fig. 3a). We first analyzed data for Winogradsky columns, which are a type of microcosm constructed by placing water and sediment, enriched with a carbon source, into a sealed transparent container and allowing it to develop under controlled lighting and temperature conditions (16). As far as we know, no published studies have reported 16S rRNA gene sequences and redox potential measurements along depth profiles in the same Winogradsky columns, so we used data from different sources to represent general features.

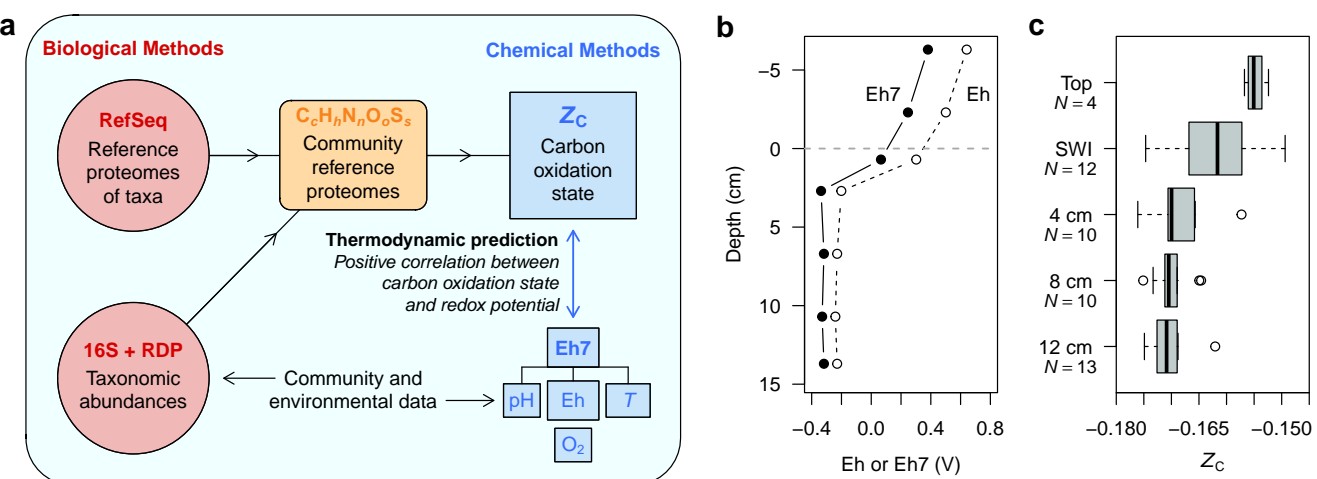

**FIG 3** Overview of methods and chemical depth profiles in different Winogradsky columns. (a) Schematic overview of data and methods used in this study. (b) Measurements of oxidation–reduction potential (denoted here as Eh; dashed line) in a Winogradsky column made with acidic sediment, taken from Fig. S5 of Diez-Ercilla et al. (17) with the depth scale adjusted so the sediment–water interface is at 0 cm. Values of Eh7 computed from Equation 1 are also shown (solid line). (c) Values of $Z_C$ for community reference proteomes computed in this study using 16S rRNA gene sequences reported by Rundell et al. (16) for Winogradsky columns composed of non-acidic sediment from ponds in Massachusetts, USA (BioProject PRJNA234104). The box-and-whisker plot represents data for samples collected from the same depth intervals in different columns. At each depth, N is the number of samples, and the center line, box width, whiskers, and points denote the median, interquartile range (IQR), the most extreme values within 1.5× IQR, and values outside this range. "Top" indicates samples collected by scraping the biofilm on the top surface of the sediment and "SWI" indicates samples of the sediment–water interface collected by drilling into the side of the column (16).

Figure 3b shows Eh measurements taken from reference (17) for a column consisting of acidic sediment and water from pit lakes in the Iberian Pyrite Belt, Spain. Because of the acidic conditions in these experiments, values of Eh7 calculated with Equation 1 are shifted to lower values than the reported redox potential, but they still exhibit a decreasing trend with depth. Redox potential also decreases with depth in the sediment of mature Winogradsky columns prepared using sediment from non-acidic ponds (108, 109), and sediments have been reported to be more reducing than the overlying water (see Fig. 5 in reference [110] and Fig. S1A in reference 111). It follows that a decrease in redox potential with depth is a general feature of mature Winogradsky columns.

In Winogradsky columns made with non-acidic sediment from ponds in Massachusetts, USA, relatively high abundances of *Alphaproteobacteria* and *Betaproteobacteria* occur near the tops of the columns, and more abundant *Firmicutes*, particularly *Clostridia*, are found in the lower parts (16). *Proteobacteria* and *Clostridia* are lineages with relatively oxidized and reduced reference proteomes, respectively (5), and changes in the relative abundances of members of these taxa are the primary drivers for the decreasing trend of $Z_C$ with depth shown in Fig. 3c. In parallel with the relatively low similarity of communities in replicate samples for the SWI as revealed by UniFrac distances (16), the SWI has the greatest variability of $Z_C$, which further illustrates that $Z_C$ is a chemical representation of the taxonomic composition of communities. Although the plots in Fig. 3b and c may be visually compelling, scatterplots are needed to make direct comparisons of $Z_C$ and Eh7 rather than comparing each with a third variable (e.g., depth). We did not make a scatterplot for Winogradsky columns because of the lack of paired redox potential and 16S rRNA data for the same columns.

## Local-scale analysis

We analyzed 68 data sets compiled from 66 studies that reported both environmental 16S rRNA gene sequences and redox potential measurements for samples from environments categorized here as river & seawater, lake & pond, geothermal areas, hyperalkaline fluids, groundwater, sediment, and soil. The data sets in our compilation have a worldwide distribution (Fig. 4a). Compared to the ranges of natural environments described by Baas Becking et al. (90), our analysis includes samples with higher pH in hyperalkaline fluids, lower Eh at near-neutral pH in some soils and freshwater systems, and lower Eh at acidic conditions in some geothermal areas (Fig. 4b). For each data set, Eh7 and $Z_C$ of the bacterial community reference proteome (referred to hereafter as bacterial $Z_C$) were used to make a scatterplot (Fig. S1). The plot legends include the number of samples (*N*), Pearson correlation coefficient (*r*), and slope (*m*). The units of slope (1/V) come from the dimensionless *y* variable ($Z_C$ has no units because it is a ratio of elemental abundances) divided by the *x* variable with units of V. If sufficient numbers of taxonomically classified and mappable archaeal sequences were available (see Materials and Methods), a second plot for archaeal $Z_C$ was also made. The selected plots in Fig. 5a show that geographically separated sediment samples from the Bay of Biscay, Spain (65), and sediment and soil samples in Hunan Province, China (71), exhibit positive correlations between Eh7 and bacterial $Z_C$, but samples from different depths of sediment cores in Daya Bay, China (60), have a negative correlation.

The correlations between Eh7 and bacterial $Z_C$ are positive for each of the five data sets for geothermal areas and have some of the highest values of slope for all environments (Fig. 5b), but the ranges of $Z_C$ differ widely depending on fluid and sample characteristics (Fig. 5c). Circumneutral to alkaline water samples from hot springs in the Eastern Tibetan Plateau (36) and New Zealand (15) have the lowest ranges of bacterial $Z_C$ not only for hot springs but also for all data sets in our compilation. Archaeal $Z_C$ exhibits both positive and negative correlations with Eh7 in geothermal areas (Fig. 5c), so the hypothesized thermodynamic effect on chemical differences appears to be stronger for the bacterial domain.

After geothermal areas, soils have the next highest frequency of positive correlations (10 out of 12 data sets or 83%). We analyzed seven soil data sets from laboratory

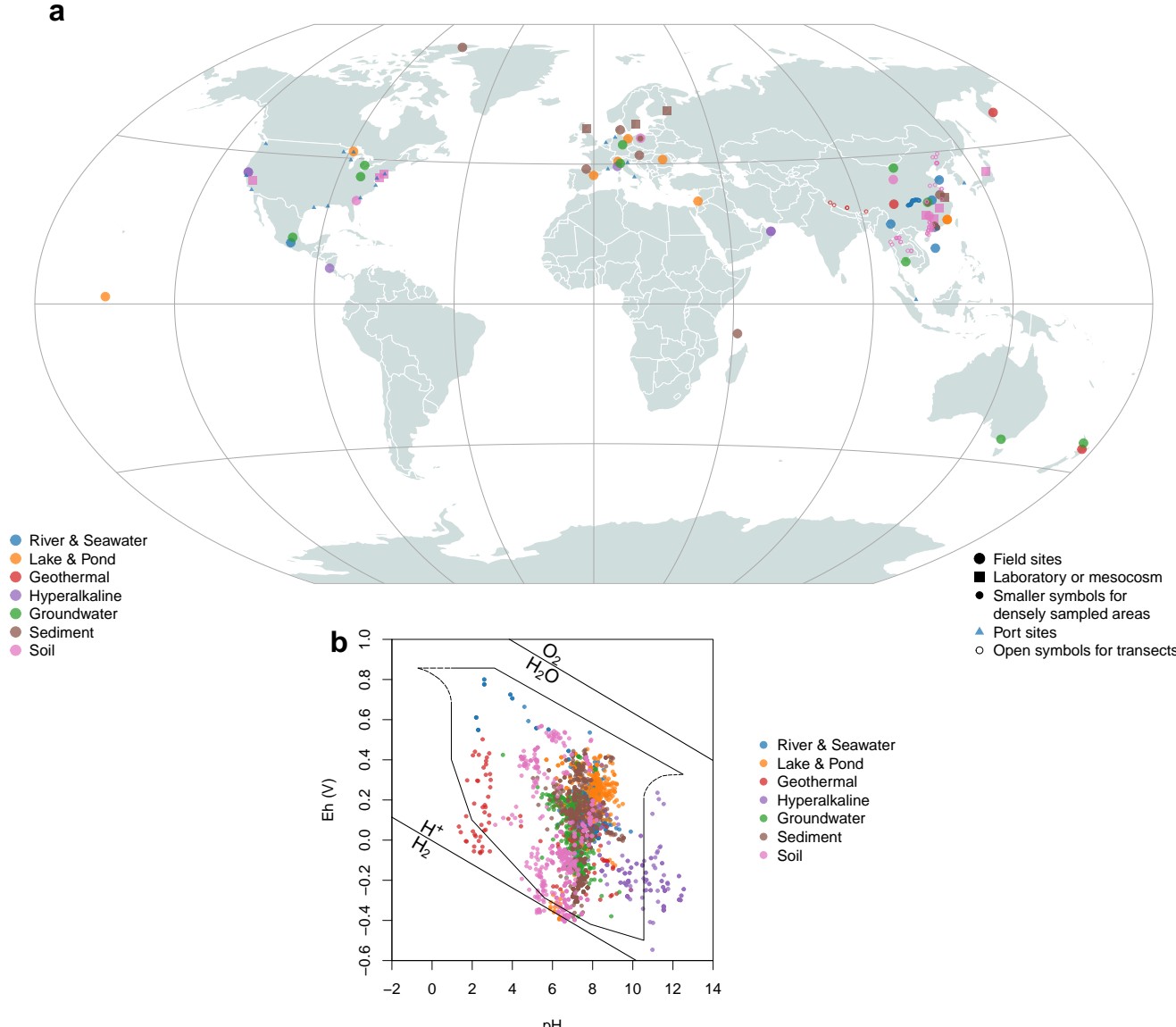

**FIG 4** Sample locations and Eh–pH diagram. (a) Sample locations were obtained from NCBI BioSample metadata or primary publications. If the location of source material for laboratory or mesocosm studies was not specified, the institutional address was used. Small open circles indicate sampling transects for paddy soils in East Asia (76), hot springs in the Southern Tibetan Plateau (38), and water samples from the Three Gorges Reservoir (25); small filled triangles represent 20 worldwide ports (14). (b) Eh–pH diagram for all environment types. The outline is redrawn from Baas Becking et al. (90) and represents the range of natural environments described in that study. Note that this figure shows values of Eh, not Eh7 as in the following figures.

or mesocosm studies in which treatments included water management (flooding and draining), amendment with organic or silicon-rich compounds, and anaerobic soil disinfestation; all of these data sets yield a positive correlation between Eh7 and bacterial $Z_C$ (see Table S1). The analysis also includes a data set for millimeter-scale variations of soil communities below the soil–water interface (83); because the sequences in this data set were obtained from environmental 16S rRNA rather than 16S rDNA, the carbon oxidation state of reference proteomes of active and not only present community members is evidently aligned with the redox gradient.

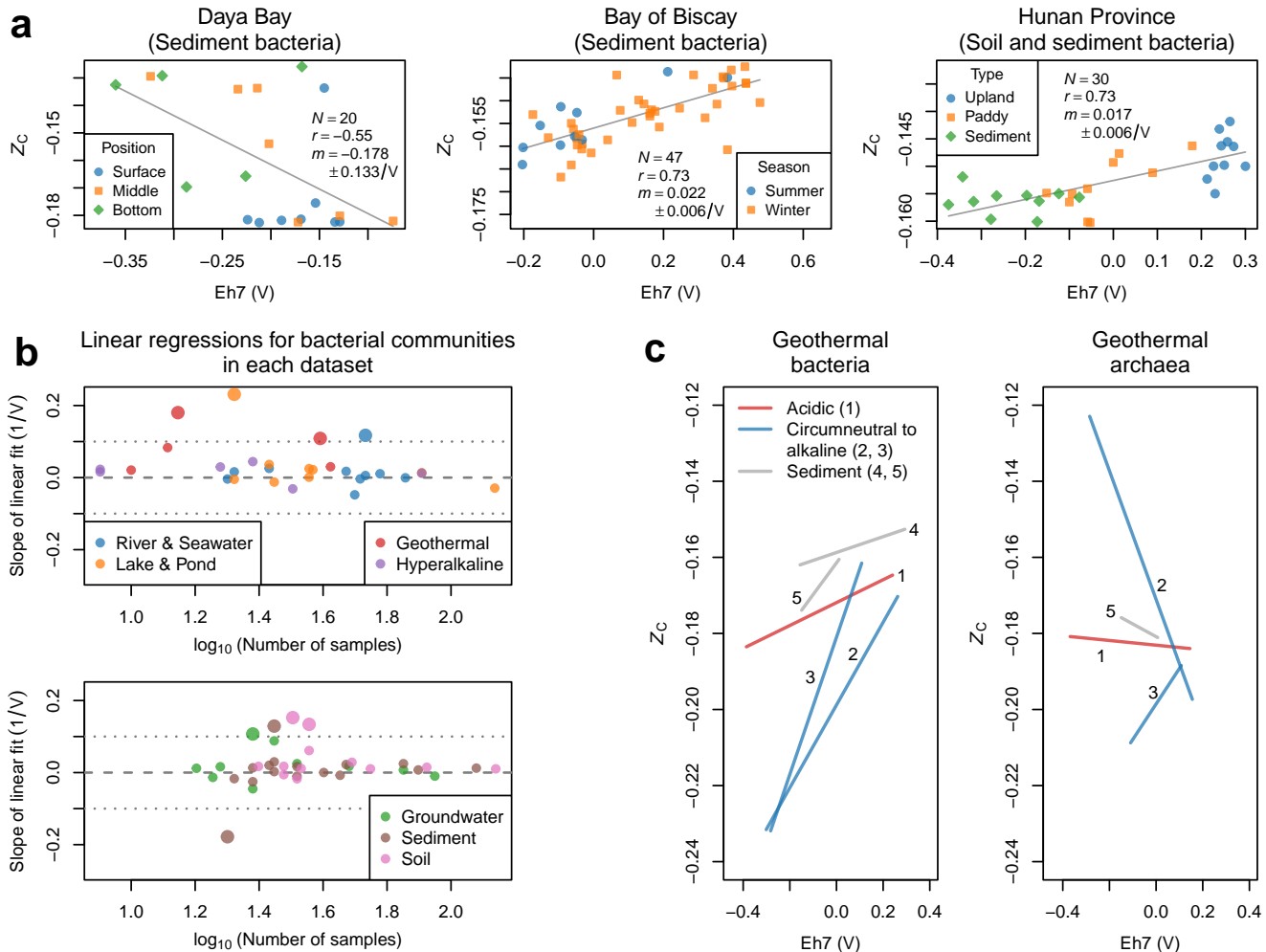

**FIG 5** Associations between Eh7 and $Z_C$ at local scales. (a) Selected data sets for sediments and soil. Scatterplots between Eh7 and bacterial $Z_C$ are shown with linear regressions. Legends indicate the number of samples ($N$), Pearson correlation coefficient ($r$), and slope of the linear regression ($m$) ± the margin of error for the 95% confidence interval. (b) Slopes of linear regressions plotted against the decimal logarithm of numbers of samples in each data set for bacterial communities. The data sets with the largest effect sizes (i.e., those for which the absolute value of slope is > 0.1/V) are indicated by larger points. Of these nine data sets, eight have positive slopes; the only one with a negative slope is the data set for Daya Bay shown in (a). (c) Linear regressions for bacterial and archaeal communities in geothermal areas. Colors are used to represent sample characteristics (acidic water, circumneutral to alkaline water, and sediment). Numbers for data sets are (1) acidic and (2) circumneutral to alkaline New Zealand hot springs (15), (3) Eastern Tibetan Plateau (36), (4) Uzon Caldera (including nine samples for acidic water and sediment and one high-pH sample) (37), and (5) Southern Tibetan Plateau (38).

## Global-scale analysis

A global analysis performed by combining individual data sets yields a positive slope for bacterial communities in each of the seven environment types (Fig. 6a). The data sets for groundwater show the strongest global correlation ($r = 0.43$), followed by lake & pond and soil (both with $r = 0.37$). Although geothermal areas exhibit strong positive correlations for bacterial communities at local scales (Fig. 5c), the differences between alkaline water and acidic water and sediments blur the pattern at a global scale. Owing at least in part to the aggregation of local-scale data sets to make the global compilation, there is a relatively large amount of scatter in the global comparison for each environment type, and it would be unwise to try to predict $Z_C$ for any particular community from the global correlation with redox potential. In contrast to bacteria, archaea show a negative global correlation in each environment type. Combining data for all environments yields positive and slightly negative correlations between Eh7 and $Z_C$ for bacteria and archaea, respectively, but only the correlation for bacteria is statistically significant

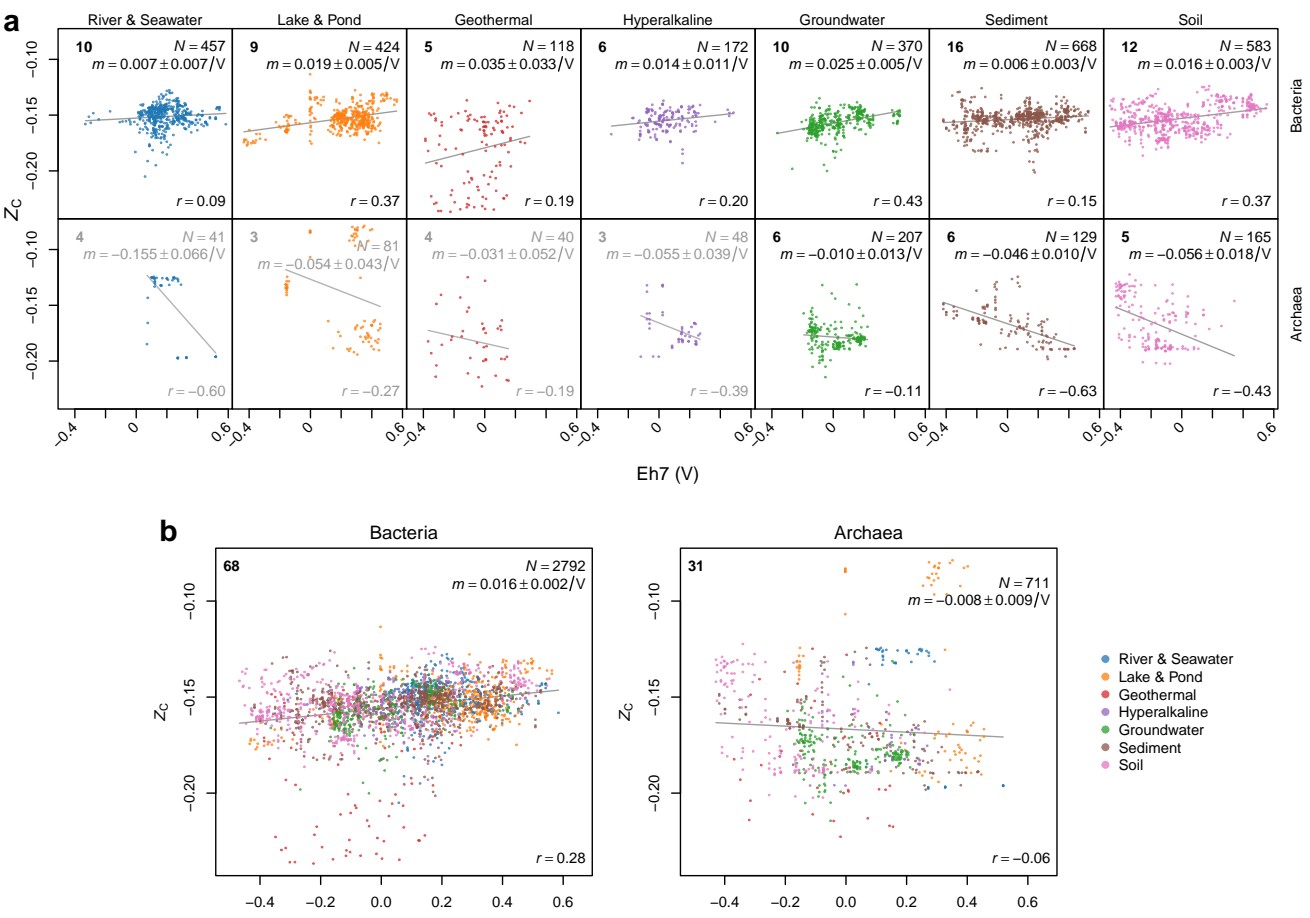

**FIG 6** Global-scale associations between Eh7 and $Z_C$. (a) Sample values and linear regressions for bacterial and archaeal communities in each environment type. The legend text is grayed out for environments represented by fewer than five data sets with archaeal sequences analyzed in this study. (b) Pan-environmental comparison. The legends indicate the number of data sets (top left), number of samples and slope of the linear regression ± the margin of error for the 95% confidence interval (top right), and Pearson correlation coefficient (bottom right).

(i.e., the 95% confidence interval of the slope for bacteria is entirely greater than zero, while that for archaea crosses zero; Fig. 6b).

Our compilation of metadata includes measurements of oxygen concentration, if available, for each data set. $O_2$ concentrations reported as zero or below the detection limit were set to zero; samples without reported $O_2$ measurements were not included in the analysis described next. Although a positive correlation between bacterial $Z_C$ and $O_2$ is evident in Fig. S2 ($N = 1,493$, $r = 0.17$), it is weaker than the correlation between $Z_C$ and Eh7 for the same set of samples ($r = 0.37$). Furthermore, bacterial $Z_C$ is more strongly correlated with Eh7 than with values of Eh that have not been corrected to pH 7 ($r = 0.29$). For the 256 samples in our compilation with both $O_2$ measurements and archaeal 16S rRNA gene sequences, the correlation with $Z_C$ is significant and positive for both Eh7 ($r = 0.17$) and $O_2$ ($r = 0.44$). This suggests that geochemical influences on archaeal protein evolution may be more strongly mediated by oxygen (explaining, in part, the lack of correlation for archaea in Fig. 6b), in contrast to an apparently stronger influence on bacterial proteins by redox potential.

## Analysis of data for one primer set

All primer sets for the 16S rRNA gene exhibit amplification bias, which refers to the variable efficiency of amplification of DNA from different taxa (112), and our global-scale analysis could in principle be affected by primer set–specific differences of amplification

bias. To control for differences of amplification bias among primer sets, we performed a second global analysis including only data sets that were generated using the 515F/806R primer set adopted for use by the Earth Microbiome Project (113) (see Table S1). This analysis reveals positive correlations between Eh7 and bacterial $Z_C$ in all environment types except river & seawater and lake & pond (Fig. S3a). Furthermore, the correlation for bacteria in the pan-environmental comparison is stronger when considering only the 515F/806R primer set ($N = 1074$, $r = 0.32$, slope = $0.023 \pm 0.004$/V; Fig. S3b) than mixed primer sets ($N = 2792$, $r = 0.28$, slope = $0.016 \pm 0.002$/V; Fig. 6b). Therefore, limiting the analysis to data sets generated using one primer set supports our main conclusions.

## Statistical analysis

Analyses of multiple data sets represent independent tests of a single hypothesis, so we used binomial probabilities to simulate the number of successes from a certain number of trials. The frequency of observed positive correlations at the local scale for bacterial communities in soil and geothermal environments could occur by chance with probabilities (binomial $P$-values) of 0.019 and 0.031, respectively, and the combined outcome for all environments has binomial $P = 0.00007$ (Table 2a). If the local-scale results for bacterial communities are filtered to include only slopes that are significantly negative or positive (i.e., those for which the minimum and maximum values in the 95% confidence interval have the same sign), then only five data sets have significantly negative slopes (two for lake & pond and three for sediment environments), and the occurrence of 31 significantly positive correlations from 36 data sets for all environments has binomial $P = 0.00001$ (Table 2b).

A 95% confidence interval that is entirely greater than zero indicates that the slopes in the global analysis are significantly positive for each environment type except for river & seawater (Fig. 6a). The correlation for river & seawater can be considered borderline significant because the 95% confidence interval of the slope extends down to but does not cross zero. Collectively, the occurrence of six or more significantly positive

**TABLE 2** Tally of regression slopes for local-scale correlations and probabilities that the results could occur by chance ($P$-values)[a]

| a | Bacteria | | | | Archaea | | | |
|---|---|---|---|---|---|---|---|---|
| | $N_{tot}$ | $N_{pos}$ | $N_{neg}$ | $P$ | $N_{tot}$ | $N_{pos}$ | $N_{neg}$ | $P$ |
| River & seawater | 10 | 6 | 4 | 0.377 | 4 | 4 | 0 | 0.062 |
| Lake & pond | 9 | 6 | 3 | 0.254 | 3 | 2 | 1 | 0.5 |
| Geothermal | 5 | 5 | 0 | 0.031 | 4 | 1 | 3 | 0.938 |
| Hyperalkaline | 6 | 5 | 1 | 0.109 | 3 | 2 | 1 | 0.5 |
| Groundwater | 10 | 7 | 3 | 0.172 | 6 | 2 | 4 | 0.891 |
| Sediment | 16 | 11 | 5 | 0.105 | 6 | 2 | 4 | 0.891 |
| Soil | 12 | 10 | 2 | 0.019 | 5 | 2 | 3 | 0.812 |
| Total | 68 | 50 | 18 | 0.00007 | 31 | 15 | 16 | 0.640 |

| b | Bacteria | | | | Archaea | | | |
|---|---|---|---|---|---|---|---|---|
| | $N_{tot}$ | $N_{pos}$ | $N_{neg}$ | $P$ | $N_{tot}$ | $N_{pos}$ | $N_{neg}$ | $P$ |
| River & seawater | 3 | 3 | 0 | 0.125 | 2 | 2 | 0 | 0.25 |
| Lake & pond | 5 | 3 | 2 | 0.5 | 1 | 1 | 0 | 0.5 |
| Geothermal | 3 | 3 | 0 | 0.125 | 0 | 0 | 0 | NA |
| Hyperalkaline | 3 | 3 | 0 | 0.125 | 3 | 2 | 1 | 0.5 |
| Groundwater | 3 | 3 | 0 | 0.125 | 2 | 0 | 2 | 1 |
| Sediment | 10 | 7 | 3 | 0.172 | 3 | 0 | 3 | 1 |
| Soil | 9 | 9 | 0 | 0.002 | 3 | 1 | 2 | 0.875 |
| Total | 36 | 31 | 5 | 0.00001 | 14 | 6 | 8 | 0.788 |

$N_{neg}$, number of data sets with negative regression slope; $N_{pos}$, of data sets with positive regression slope; $N_{tot}$, total number of data sets analyzed;.
[a](a) All data sets; (b) data sets filtered to include only those with statistically significant slopes. $P$ is the probability for $N_{pos}$ or more successes to occur by chance in $N_{tot}$ trials with a 0.5 hypothesized probability of success in each trial.

correlations for bacterial communities in seven environment types has binomial $P$ = 0.0625. Furthermore, of the nine local-scale data sets for which the absolute value of the slope is greater than 0.1/V, eight have positive slopes (Fig. 5b). In other words, a predominance of positive slopes is exhibited by the subset of data sets with the largest effect size and not only by all data sets in the compilation for bacterial communities.

## DISCUSSION

This study is primarily a contribution toward geochemical biology, which is a research area concerned with the effects of geochemistry on genome evolution. To test the hypothesis that protein sequences coded by genomes in microbial communities are chemically adapted to redox conditions, we assessed the sign of correlations between $Z_C$ of community reference proteomes and environmental redox potential corrected to pH 7 for more than 60 publicly available data sets. We observed positive correlations for a majority of data sets for bacterial communities at local scales (Table 2) and at a global scale for each of seven environment types (Fig. 6). Despite this, the regressed slopes vary widely (Fig. 5b), and there is a substantial spread around the regression line in most scatterplots. Therefore, redox conditions appear to have a modulating rather than controlling effect on the carbon oxidation state of bacterial protein sequences.

A limitation of this study is that reference proteomes carry no information about protein abundance and cannot be used to make inferences about the effects of geochemistry on protein expression levels or whole-cell elemental composition. As proteins typically make up approximately half of the cellular dry weight, the elemental composition of cells is strongly related to protein expression. On the one hand, highly expressed proteins require fewer total mutations than lowly expressed proteins for stoichiometric constraints to be accommodated by sequence evolution. On the other hand, lowly expressed proteins evolve faster than highly expressed proteins (114). A tradeoff between abundance and evolvability suggests a reason for the sequences of both highly and lowly expressed proteins to be chemically adapted to environmental conditions and may explain in part why community reference proteomes exhibit correlations with redox conditions despite having assumed equal contributions from proteins that actually vary widely in abundance.

Biological and technical biases represent additional limitations that demand a conditional rather than definitive conclusion. Horizontal gene transfer and large mobile genetic elements (e.g., plasmids) are not adequately represented by reference proteomes, and there are multiple sources of bias that affect the analysis of gene sequence data sets–from DNA extraction to sequencing to bioinformatic processing. These biases affect comparisons both within and between data sets; they do not cancel out even for the same sequencing method. Therefore, the findings of this study–and many others–are potentially invalid (115). Bias, instead of true biological variation, could explain why correlations for some data sets are stronger than others. Confirming these trends by using another method (e.g., shotgun metagenomes) would help give more confidence to these results, although bias is also problematic for shotgun metagenomic sequencing (115) and for metaproteomics analysis (116).

Among data sets for extreme environments, hypersaline lakes in the Monegros Desert of Spain (34) are characterized by very high $Z_C$ values of the archaeal communities (see Fig. S1 and the plot for lake & pond data sets in Fig. 6a). This trend is consistent with highly oxidized proteins inferred from metagenomes in other hypersaline samples (12). In contrast, moderately alkaline hot springs have the lowest range of $Z_C$ for all environments considered here. The consistently positive correlations for geothermal areas despite differences in pH and sample type (Fig. 5c) suggest that bacterial metacommunities can attain a low-energy state with respect to redox gradients in a complex physicochemical milieu. However, redox potential does not predict the large offsets of $Z_C$ among circumneutral to alkaline water, acidic water, and geothermally heated sediment samples, so other chemical and biological factors probably drive the divergent evolution that is apparent in different types of hot spring samples.

There is a striking contrast between negative correlations observed for some sediment cores (see Daya Bay in Fig. 5a and Mai Po Wetland and Lake Neusiedl in Fig. S1) and the mostly positive correlations for soil data sets. Soils exhibit low phylogenetic diversity despite high species-level diversity; in contrast, sediments have higher phylogenetic diversity than many other environments (117). These differences might imply a greater extent of redundancy in the elemental composition of reference proteomes for distinct species in soils, which could contribute to the higher frequency of positive correlations between Eh7 and bacterial $Z_C$ for soils compared to sediments. At a global scale, it is not sediments but river & seawater communities that exhibit the weakest correlation compared to other environments. In contrast to lakes, many of which exhibit vertical redox stratification, the distribution of Eh7 for river & seawater data sets is weighted toward mostly positive values (Fig. 6a), suggesting that the magnitude of the sampled redox gradients is less in those systems; this may reduce the power to detect an association, if there is one, between Eh7 and $Z_C$.

In the global pan-environmental comparison, the carbon oxidation state of reference proteomes for bacterial communities is more strongly associated with Eh7 than either Eh or $O_2$ concentration. A significant association with Eh7 is absent from the global pan-environmental comparison for archaea (Fig. 6b), but limiting the analysis to samples with reported oxygen concentrations reveals a positive association for archaea that is stronger for $O_2$ than for Eh7, in contrast to bacteria (Fig. S2). This could suggest that protein evolution in archaea is more sensitive to oxygen levels, whereas bacterial protein sequences may be stronger indicators of redox potential than oxygen. However, we consider the results for archaea to be more uncertain because of limitations of archaea-specific primers (e.g., see reference 32) and the absence from the RefSeq database of representatives of the DPANN superphylum such as Woesearchaeota and Pacearchaeota that are abundant in some environments (118). Across all data sets, the average percentages of genus-level classifications made by the RDP Classifier that were mapped to the NCBI taxonomy are 86% for bacteria and 77% for archaea, suggesting that archaeal communities are not as well represented by available reference proteomes. In addition, reference proteomes for archaea are likely to be lower quality because automated annotation pipelines may not be current with recently discovered translation mechanisms (119).

## Conclusions

In the context of millions of years of sequence evolution, a geochemical thermodynamic model makes a directional prediction about the differences of elemental composition of protein sequences along environmental redox gradients. We found initial support for this prediction by observing depth-wise decreases of redox potential and carbon oxidation state of community reference proteomes from data reported for different Winogradsky columns. Our main analysis of 68 data sets for bacterial communities uncovered predominantly positive $Z_C$–Eh7 correlations at local scales and a positive correlation globally in each of the seven environment types. These results suggest that chemical differences between protein sequences occurring at the millimeter scale or 1,000-km scale may, to some extent, be explained by a single hypothesis of energy minimization shaping protein evolution.

Our conclusions are based on protein sequences from reference genomes and do not account for protein abundances. We could not find metaproteomic data sets that would permit comparing protein abundances with Eh measurements. Analysis of data from selected studies that reported both metaproteomes and 16S rRNA gene sequences revealed a positive but weak correlation between metaproteomic and 16S rRNA-based estimates of carbon oxidation state. The higher values for metaproteomes appear to be consistent with higher expression of relatively oxidized proteins, but we cannot rule out that the differences may be due to technical bias.

Our findings imply that redox gradients may be major chemical drivers of community structure, in addition to previously recognized physicochemical factors including

temperature and pH (120). Because of the negative or non-existent association between $Z_C$ and Eh7 in some individual data sets and the scatter in the global-scale analysis, we infer that geochemistry modulates, rather than controls, the underlying evolutionary and ecological processes that alter genome sequences and relative abundances of taxa. Based on these findings, it can be expected that redox potential is one of many factors that influence the elemental composition of bacterial communities, but elemental analysis of biomass, amino acid analysis of bulk protein, or measurements of metaproteomic abundances are needed to more directly test that prediction. Experimental evolution under defined redox conditions, such as the evolution of *E. coli* under aerobic or anaerobic conditions (121), could provide experimental tests of the chemical adaptation hypothesis.

## ACKNOWLEDGMENTS

We are grateful to Matti Ruuskanen for providing mapping files to demultiplex the sequence data for Lake Hazen.

D.M. acknowledges funding from the Natural Science Foundation of Changsha (kq2202089) and the Key Research and Development Program of Hunan Province (2022SK2076 and 2020WK2022).

J.M.D. conceived the idea for the study. D.M. provided data. J.M.D. wrote the paper with input from D.M. Both authors participated in revising the paper and interpreting results.

We declare that we have no conflicts of interest.

## AUTHOR AFFILIATIONS

[1]Key Laboratory of Metallogenic Prediction of Nonferrous Metals and Geological Environment Monitoring of Ministry of Education, School of Geosciences and Info-Physics, Central South University, Changsha, China
[2]Key Laboratory of Biometallurgy of Ministry of Education, School of Minerals Processing and Bioengineering, Central South University, Changsha, China

## AUTHOR ORCIDs

Jeffrey M. Dick ⓘD http://orcid.org/0000-0002-0687-5890

## FUNDING

| Funder | Grant(s) | Author(s) |
| --- | --- | --- |
| Key Research and Development Program of Hunan Province | 2020WK2022, 2022SK2076 | Delong Meng |
| Natural Science Foundation of Changsha | kq2202089 | Jeffrey M. Dick |

## AUTHOR CONTRIBUTIONS

Jeffrey M. Dick, Conceptualization, Data curation, Formal analysis, Investigation, Writing – original draft | Delong Meng, Funding acquisition, Investigation, Writing – review and editing

## DATA AVAILABILITY STATEMENT

The data sets supporting the conclusions of this article are available in two R packages. chem16S is developed on GitHub (https://github.com/jedick/chem16S); version 0.1.3 of the package used in this study is archived on Zenodo (122). chem16S was used for calculating chemical metrics of community reference proteomes from RDP Classifier output and includes the amino acid compositions of reference proteomes for taxa derived from RefSeq. JMDplots is developed on GitHub (https://github.com/jedick/JMDplots); version 1.2.17 of the package used in this study is archived on Zenodo (123).

JMDplots has been developed to support multiple studies; the sequence processing script, RDP Classifier output, compiled sample metadata, and code written for this study are specifically available in the orp16S section of the package, and the orp16S. Rmd vignette runs the code to make each of the figures in the paper.

## ADDITIONAL FILES

The following material is available online.

### Supplemental Material

**FIG S1 (Figure_S1.pdf).** $Z_C$–Eh7 scatterplots and linear regressions for all data sets. Subtitles indicate the number of samples, Pearson correlation coefficient, and slope of the linear regression ± the margin of error for the 95% confidence interval. Regression lines are solid if the slope is > 0.01/V or < −0.01/V, or dashed otherwise.
**FIG S2 (Figure_S2.pdf).** Comparison of Eh7, Eh, and $O_2$ concentration as predictors of carbon oxidation state. Within each domain, all plots represent the same set of samples (i.e., those for which $O_2$ measurements are available). The number of data sets is indicated by bold numbers at the top left of each plot; the number of samples and slope of the linear regression ± the margin of error for the 95% confidence interval are shown at upper right; the Pearson correlation coefficient is shown at bottom right. The numbers of samples for each environment type are listed in the bottom legends.
**FIG S3 (Figure_S3.pdf).** Global analysis including only data sets generated with 515F/806R primers (these data sets are identified in Table S1). This figure was made analogously to Fig. 6
**TABLE S1 (Table_S1.xlsx).** Data set summaries: Bibliographic key, name, 515F/806R primer set (yes/no), number of samples with bacterial and archaeal sequences, sample type (soil, sediment, water, or biological), ranges of $T$ (°C), pH, Eh (mV), Eh7 (mV), sign of $Z_C$–Eh7 correlation for bacteria, sources of Eh and $O_2$ data, notes.
**TABLE S2 (298155_1_supp_6646473_rqsb1h.xlsx).** Sequence processing statistics.

### Open Peer Review

**PEER REVIEW HISTORY (review-history.pdf).** An accounting of the reviewer feedback and comments.

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
