## [Reviewer comments · mSystems]

Community- and genome-based evidence for a shaping influence of redox potential on bacterial protein evolution

Jeffrey Dick and Delong Meng

Corresponding Author(s): Jeffrey Dick, Central South University

Review Timeline:

Submission Date:	January 7, 2023
Editorial Decision:	February 20, 2023
Revision Received:	February 28, 2023
Accepted:	February 28, 2023

Editor: Christopher Schadt

Reviewer(s): Disclosure of reviewer identity is with reference to reviewer comments included in decision letter(s). The following individuals involved in review of your submission have agreed to reveal their identity: Avi I Flamholz (Reviewer #3)

Transaction Report:

DOI: <https://doi.org/10.1128/msystems.00014-23>

February 20, 2023

Dr. Jeffrey M. Dick
Central South University
School of Geosciences and Info-Physics
Changsha
China

Re: mSystems00014-23 (Community reference proteomes suggest widespread influence of geochemistry on bacterial protein evolution)

Dear Dr. Jeffrey M. Dick:

Thank you for submitting your manuscript to mSystems. We have completed our review and I am pleased to inform you that, in principle, we expect to accept it for publication in mSystems. However, acceptance will not be final until you have adequately addressed the reviewer comments.

Preparing Revision Guidelines

Sincerely,

Christopher Schadt

Editor, mSystems

Journals Department
American Society for Microbiology

Reviewer comments:

Reviewer #1 (Comments for the Author):

I appreciate the authors' attention to reviewer comments and their attempts to address the original submission's shortcomings. The manuscript has been appropriately softened and needed information added to the documents. Below are some comments, with most of my remaining minor concerns regarding the new comparison provided in Figure 2b.

- In reference to point set 1 (Fig 2b and associated text): I appreciate the authors providing this additional comparison showing the differences between using a purely 16S/Ref proteome approach to one where metaproteomes were actually measured. As stated in line 290-291, the correlation is weakly positive. And this weak correlation is used as evidence to state (line 308-310) that the 16S/Ref proteome approach is more than sufficient. That said, I still am uncomfortable with the weak correlations here (and sometimes throughout) - but I am trying to titrate my analytical/quantitative mindset to be more "forgiving" towards broad ecological data and trends. The authors did speculate that this is due to the technical underrepresentation of membrane proteins in proteome measurements and higher copy number of cytoplasmic proteins - while plausible, this is still somewhat speculative. Further the higher copy number of cytoplasmic proteins is still a real, natural occurrence, and if these proteins play an outside role on the Zc calculation (to its correlational detriment), especially with protein abundances correctly considered, it's hard in my opinion to just discount these metrics and go all in with the 16S/Ref proteome strategy, especially as they are not weighted appropriately to resource demand in the cell (i.e. building proteins with reducing or non-reducing amino acids and to what extent). However, the authors explain that they are limited with data that's publicly available. With regard to the active chimneys, (line 316) "only the reference proteomes clearly separate them from the other datasets". I would argue, as you had in the previous clause, that they are amongst the most reduced communities being on the left edge of the distribution of Zc (x axis). Who's to say that the expressed proteomes are more or less predictive than a dimensionless 16S/Ref proteome calculation where protein expression is totally disregarded. This all said, I do appreciate the overall softening of the language, especially in the new conclusion sentence on line 634.

- Overall, the authors cleaned up their "unnecessary hedging" with the data. This provides improved clarity in the final document, but it still leaves me wondering about N= size especially with R2 values that are low from what I am used to dealing with/observing in non-ecological studies.

- The title was softened appropriately.

- They have added the necessary clarity in how they performed the proteome abundance / AA-based Zc calculation. Although spectral counting was used, as opposed to calculating peptide chromatographic area-under-the-curve (AUC), this could be a reasonable proxy. We have found AUC metrics to be more indicative of abundance, but perhaps this can be explored in another manuscript.

Reviewer #3 (Comments for the Author):

The authors, Jeffrey Dick and Delong Meng, have made substantial edits that are more than sufficient to address my concerns about clarity of the underlying hypotheses. The paper can, in my view, be accepted as is and published immediately.

However, if the authors have patience and interest, I've given some suggestions below. Some editing can make this lovely manuscript more approachable to a wide audience of microbiologists and systems biologists, which is what mSystems offers. I look forward to seeing it published and available online.

1. The new introduction addresses all my concerns about clarity, e.g. about what reference proteomes are, what intellectual exercise is being conducted. Some minor issues:

L57: phylostratigraphy is jargon that I was not familiar with. I can guess what this is, but many readers may not be up to date with ancestral sequence reconstruction methods.

L82: "possibly expressed proteins" tripped me up -- write "potentially expressed" to avoid incorrect parsing of the clause.

L77-94: really great justification of the approach. very helpful to have in the introduction.

L96-97: something is wrong here grammatically.

L98-100: explicitly state that O₂ is not the only relevant oxidant and H₂ is not the only relevant reductant. This motivates the use of an electrode, which should also be stated explicitly. Not being an expert in electrochemistry myself, I assume it matters what type of electrode one uses, i.e. I assume different organics will react with different electrode materials? Perhaps this is worth mentioning as a caveat if so.

2. The new "theoretical background" section is excellent and much appreciated.

-- Figure 1 might include some conceptual diagram, e.g. of the evolution of genomes to include more reduced amino acids in reducing environments.

-- I am not sure I understand Figure 1B. Perhaps I haven't understood how to think about Eh₇. Eh₇ is defined in the text as the Eh that would have been measured if the pH was 7, presuming a specific value for dEh/dpH. Naively it seems to me like pH should also affect the "incomplete metastable equilibrium" protein composition, but it seems from the figure that these effects are minor in principle. A fuller description here would help me and others.

-- There is also room to cut from this section, which has a fair bit of technical detail. There is currently no matching methods section. Perhaps some details (e.g. of the QEC model and CHNOSZ package) could be moved to a methods section to focus the text more on concepts like Eh₇.

L131: the comment about protein folding comes out of left field.

L137: worth explicitly saying that you assume that every coding sequence is equally important (i.e. not using expression data). I think this is equivalent to saying they have equal expression levels (even though protein expression varies over 6 orders!). After all, the reason evolution "cares" about protein Z_C is the cells need to make the proteins and (often) the constituent aminos in whatever environment they live in.

L158: least unstable protein -> least unstable reference proteome? it's clearly not just one protein, right?

L156: "relative stability fields" is hard to understand unless you are familiar with the geochemistry language. I'd guess most readers are not.

L213: why not include host-associated measurements if Eh is also reported? I don't follow. Regardless, there is enough analysis in this paper that this sentence can be cut.

L244: might be worth giving a number. how much more reduced are typical membrane proteins? I remember some discussion in Dick 2014 on this topic.

3. Statistical analysis -

L393-403 I found this paragraph somewhat repetitive with the previous.

3. Discussion.

- A major finding is that Archaea show uncertain, often negative Eh₇-Z_C correlations. One explanation for this result is that the reference proteomes for Archaea are simply lower quality than those for bacteria. Many factors can limit the automated annotation of protein sequences from closed genomes, for example annotation of transcription start sites is ambiguous, as is identification of stop-codon suppression (e.g. for selenocysteine, pyrrolysine). See this reference for an example D. R. Gelsinger, et al., *Nucleic Acids Res.* 48, 5201-5216 (2020).

- A related concern to the above. Uniprot contains reference proteomes for ≈350 Archaea and ≈8800 bacteria. Perhaps the bacteria are better represented by (more closely related to?) their best-hit reference proteome than the Archaea analyzed on average?

L417: "from thermodynamics" -> "from our thermodynamic model of relative coding sequence stability" ... there is more than one reasonable thermodynamic model here.

L446-462: I am not sure I understand whether dispersal limitation is "good" or "bad" for your hypothesis. It took me two reads to understand that you think dispersal is "good" because it allows for the well-adapted microbes to get to the right redox niche, yes? But one could also imagine that lack of dispersal gives "time" for evolutionary sequence adaptation, or preferential growth of those microbes that are well-adapted. I think it is worth being a bit more explicit and clear here.

More broadly, on the same L446-462... you convinced me that the point of this exercise is to ask whether you see a strong statistical association between coding sequence Z_C and Eh7. Figure 6 makes a good case from binomial statistics (for bacteria), but does that worldview permit such close inspection of individual datasets? Doesn't that amount to testing additional hypotheses? Your methods section claims there is only one hypothesis...

L464-479: I feel you are getting ahead of yourself speculating about Archaeal physiology given the methodological concerns related to Archaeal ref. proteomes. A good opportunity to trim the text here.

4. Conclusion does a good job of tying the paper together.

L493-497: I got a little lost here. I don't understand how metaproteomics and 16S are related exactly.

L499: Would rephrase here. It is worth being explicit that bacteria are typically $\approx 50\%$ protein by dry weight, so elemental composition is strongly related to protein expression. You made an assumption that coding sequences are a good estimate for expressed proteins. This is obviously problematic but a nice starting point. And, moreover, it gives surprising results.

L503-516: really nice closing. Honest and clear

To the Editor and Reviewers:

We have made revisions to the manuscript as described in the point-by-point response below. In addition, we felt compelled to change the title for reasons given in the Justification for Title Change section at the end of this response.

A related preprint has been posted on bioRxiv (<https://doi.org/10.1101/2023.02.12.528246>). The left-side panels of Figure 1 in that preprint present a similar analysis to Figure 2 of the current manuscript, but are based on a different reference database (GTDB instead of RefSeq) and include data for human microbiomes, which were not analyzed in the current study.

All line numbers in our response refer to the Marked-Up Manuscript.

Reviewer comments:

Reviewer #1 (Comments for the Author):

I appreciate the authors' attention to reviewer comments and their attempts to address the original submission's shortcomings. The manuscript has been appropriately softened and needed information added to the documents. Below are some comments, with most of my remaining minor concerns regarding the new comparison provided in Figure 2b.

Thank you for your comments.

• In reference to point set 1 (Fig 2b and associated text): I appreciate the authors providing this additional comparison showing the differences between using a purely 16S/Ref proteome approach to one where metaproteomes were actually measured. As stated in line 290-291, the correlation is weakly positive. And this weak correlation is used as evidence to state (line 308-310) that the 16S/Ref proteome approach is more than sufficient.

To avoid creating the impression of over-confidence, we have deleted this line: “Therefore, the use of 16S rRNA sequences and reference proteomes permits a viable first-order estimate of chemical metrics for protein sequences but not protein abundances in microbial communities.” (Line 324 in the current Marked-up Manuscript, corresponding to line 308 mentioned by the Reviewer)

That said, I still am uncomfortable with the weak correlations here (and sometimes throughout) - but I am trying to titrate my analytical/quantitative mindset to be more "forgiving" towards broad ecological data and trends.

Please see our response to the comment about “R2 values” below.

The authors did speculate that this is due to the technical underrepresentation of membrane proteins in proteome measurements and higher copy number of cytoplasmic proteins - while plausible, this is still somewhat speculative.

We borrowed the “copy number” terminology from Masuda et al. (2009), but their study is about protein abundance in cells, not copy numbers of genes in the genome. Therefore, we have changed “copy number” to “abundance”. We have also softened this sentence by adding “possibly” and “are two factors” – thereby implying that there may be other factors (Line 313):

Therefore, technical bias against detecting membrane proteins in MS-based proteomics and the higher natural **abundance** of cytoplasmic than membrane proteins **are two factors that** could

possibly contribute to the higher Z_C of proteins observed in metaproteomes compared to community reference proteomes.

Further the higher copy number of cytoplasmic proteins is still a real, natural occurrence, and if these proteins play an outsize role on the Z_C calculation (to its correlational detriment), especially with protein abundances correctly considered, it's hard in my opinion to just discount these metrics and go all in with the 16S/Ref proteome strategy, especially as they are not weighted appropriately to resource demand in the cell (i.e. building proteins with reducing or non-reducing amino acids and to what extent). However, the authors explain that they are limited with data that's publicly available.

We appreciate the Reviewer's concern for an accurate representation of protein abundances in cells. As mentioned in the Introduction, comparison of genome sequences (and, by extension, reference proteomes) allows focusing the analysis on evolutionary rather than physiological differences (Line 113). Please see also our response to Reviewer #3's comment about "After all, the reason evolution "cares" ...".

One possibility for future refinement of this technique would be to partition the reference proteomes – for instance, instead of using all proteins in each genome, analyze the elemental compositions only of ribosomal proteins, housekeeping proteins, or other families that are highly abundant and/or localized to the same subcellular compartment; this could partially control for different abundances of cytoplasmic and membrane proteins. The analysis of metagenomic data by Dick and Shock (2011) showed that Z_C of proteins with different functional annotations are offset from one another but have parallel trends along an environmental redox gradient. The names in this plot come from automatic annotation of metagenomic protein sequences, and the sites are on an environmental gradient from hot and reducing (1) to cool and oxidizing (5). Note the relatively low Z_C of membrane proteins and permeases, which include transporters with membrane-spanning domains.

In the context of the Reviewer's comment, is notable that "overall", which combines all protein sequences together (the metagenomic analog to community reference proteomes), has nearly the same slope as individual functional categories.

With regard to the active chimneys, (line 316) "only the reference proteomes clearly separate them from the other datasets". I would argue, as you had in the previous clause, that they are amongst the most reduced communities being on the left edge of the distribution of Z_C (x

axis). Who's to say that the expressed proteomes are more or less predictive than a dimensionless 16S/Ref proteome calculation where protein expression is totally disregarded.

We revised these lines as follows (Line 329):

This result shows that the highly reducing environment associated with hydrothermal fluids corresponds to highly reduced community reference proteomes. Notably, metaproteomes indicate that the expressed proteins of communities in active chimneys are also relatively reduced compared to those in most other environments. Unfortunately, metaproteomic data were not available for assessing correlations with redox potential in the remainder of this study.

This all said, I do appreciate the overall softening of the language, especially in the new conclusion sentence on line 634.

Thank you. Please note that this sentence has been revised and split into two sentences (Line 628):

Analysis of data from selected studies that reported both metaproteomes and 16S rRNA gene sequences revealed a positive but weak correlation between metaproteomic and 16S rRNA-based estimates of carbon oxidation state. The higher values for metaproteomes appear to be consistent with higher expression of relatively oxidized proteins, but we cannot rule out that the differences may be due to technical bias.

• Overall, the authors cleaned up their "unnecessary hedging" with the data. This provides improved clarity in the final document, but it still leaves me wondering about N= size especially with R2 values that are low from what I am used to dealing with/observing in non-ecological studies.

The strength of correlations is similar to other ecological studies. A direct comparison is not possible because no previous studies have used the same variables, but here is an example from a global soil microbiome study (Figure 1 of Bahram et al., 2018).

Those authors noted that the correlation for taxonomic diversity is relatively weak. For comparison with our results, panel (a) may be most appropriate because our analysis was based in part on taxonomic abundances of bacteria. The correlation for global soil datasets in our study has $r = 0.37$, corresponding to $r^2 = 0.14$.

- **The title was softened appropriately.**

Please note that the title has been changed. The word “suggests” has been removed; however, the phrase “widespread influence” has been replaced by “shaping influence” to keep a soft tone.

Precedent for a similar usage of “shape” comes from a recent article in *mSystems* entitled “Environmental selection and biogeography shape the microbiome of subsurface petroleum reservoirs” (Gittins et al., 2023). A plot from that paper illustrates the level of variability that can be present in a global-scale microbial diversity analysis, yet the overall distribution reflects a shaping influence. Here, Bray-Curtis dissimilarity represents taxonomic differences between communities (beta diversity).

- **They have added the necessary clarity in how they performed the proteome abundance / AA-based Zc calculation. Although spectral counting was used, as opposed to calculating peptide chromatographic area-under-the-curve (AUC), this could be a reasonable proxy. We have found AUC metrics to be more indicative of abundance, but perhaps this can be explored in another manuscript.**

Thank you for this comment.

Reviewer #3 (Comments for the Author):

The authors, Jeffrey Dick and Delong Meng, have made substantial edits that are more than sufficient to address my concerns about clarity of the underlying hypotheses. The paper can, in my view, be accepted as is and published immediately.

However, if the authors have patience and interest, I've given some suggestions below. Some editing can make this lovely manuscript more approachable to a wide audience of microbiologists and systems biologists, which is what *mSystems* offers. I look forward to seeing it published and available online.

Thank you for your comments.

- 1. The new introduction addresses all my concerns about clarity, e.g. about what reference proteomes are, what intellectual exercise is being conducted. Some minor issues:**

L57: phylostratigraphy is jargon that I was not familiar with. I can guess what this is, but many readers may not be up to date with ancestral sequence reconstruction methods.

Added “a technique in which conservation levels of orthologous genes among species are used to estimate ages of gene families”. (Line 71)

L82: "possibly expressed proteins" tripped me up -- write "potentially expressed" to avoid incorrect parsing of the clause.

Changed as suggested. (Line 111)

L77-94: really great justification of the approach. very helpful to have in the introduction.

L96-97: something is wrong here grammatically.

Reworded as: “Concentrations of dissolved oxygen (O₂) diminish to levels that are below detection in many sediments (e.g., within several millimeters to centimeters of the sediment-water interface; see reference 6) and other reducing environments.” (Line 134)

L98-100: explicitly state that O₂ is not the only relevant oxidant and H₂ is not the only relevant reductant. This motivates the use of an electrode, which should also be stated explicitly.

These lines have been added (Line 140):

Oxygen and hydrogen, and other relevant oxidants and reductants, do not attain equilibrium in most environments. Nevertheless, a single redox scale is needed for a global comparison of microbial habitats. Furthermore, the redox property should be detectable in all environments; neither O₂ nor H₂ concentration satisfies this criterion, but electrode measurements do.

Not being an expert in electrochemistry myself, I assume it matters what type of electrode one uses, i.e. I assume different organics will react with different electrode materials? Perhaps this is worth mentioning as a caveat if so.

As mentioned in the text, a platinum indicator electrode is used for Eh measurements (Line 150). The electroactive species (i.e., those that react with the electrode on the timescale of the measurements) in natural systems are often not well defined. The sentence has been revised as follows (Line 154):

A rigorous chemical interpretation of Eh measurements in natural systems is challenging because they represent mixed potentials that are affected by the presence of various, often not well defined, electroactive species that are generally not in mutual equilibrium (Bricker, 1982).

2. The new "theoretical background" section is excellent and much appreciated.

Thank you.

-- Figure 1 might include some conceptual diagram, e.g. of the evolution of genomes to include more reduced amino acids in reducing environments.

Agreed, a conceptual diagram would be nice. Maybe it could be done (for a later paper) with a conceptual drawing of an evolutionary tree, with different tips of the tree placed under reducing and oxidizing sites.

-- I am not sure I understand Figure 1B. Perhaps I haven't understood how to think about Eh7. Eh7 is defined in the text as the Eh that would have been measured if the pH was 7, presuming a specific value for dEh/dpH. Naively it seems to me like pH should also affect the "incomplete metastable equilibrium" protein composition, but it seems from the figure that these effects are minor in principle. A fuller description here would help me and others.

Added to caption of Figure 1 (p 5, no line number):

The upper plot shows Z_C of the most stable reference proteome at three pH values, corresponding to the vertical dashed lines in (a). Note, for instance, that the reference proteome for Mvo, which is relatively stable at pH = 5, has Z_C that is intermediate between those of Mma and Msm, which are relatively stable at pH = 9. The lower plot shows the same profiles after correction to Eh7 using Equation 1.

Added to text (Line 230):

It should be noted that the relative stabilities of proteomes depend on both pH and Eh; the correction to Eh7 is independent from the metastable equilibrium calculation and is only used to illustrate how values of Eh obtained at different pH values can be compared.

-- There is also room to cut from this section, which has a fair bit of technical detail. There is currently no matching methods section. Perhaps some details (e.g. of the QEC model and CHNOSZ package) could be moved to a methods section to focus the text more on concepts like Eh7.

Changed as suggested. Details of the thermodynamic calculations have been moved to Materials and Methods. (Line 659)

L131: the comment about protein folding comes out of left field.

We have deleted this sentence as well as the reference to protein unfolding in the caption of Figure 1. (Line 174 and p 5, no line number)

L137: worth explicitly saying that you assume that every coding sequence is equally important (i.e. not using expression data). I think this is equivalent to saying they have equal expression levels (even though protein expression varies over 6 orders!).

L137 in the previous clean manuscript corresponds to Line 182 in the current marked-up manuscript, and L137 in the previous marked-up manuscript corresponds to Line 109 in the current marked-up manuscript. We are not certain that either of these locations is the best place for this information. Instead, we have added some lines to the Introduction (Line 93):

Actual protein abundances are not available in RefSeq and were not used in our study. Instead, for each species, a reference proteome was obtained by summing the amino acid compositions of all coding sequences in the genome; that is, each sequence was assumed to be equally important. Then, species reference proteomes were aggregated to higher taxonomic levels and finally multiplied by taxonomic abundances to obtain community reference proteomes.

After all, the reason evolution "cares" about protein Z_C is the cells need to make the proteins and (often) the constituent aminos in whatever environment they live in.

In terms of energy budgets, highly expressed proteins are the most important. However, lowly expressed proteins are more evolvable; see Bédard et al. (2022) and our response to "Would rephrase here" further below. Given the dearth of information about metaproteomic abundances in our chosen environmental context (i.e., where Eh measurements have been reported), there is room to glean insight into evolutionary differences from the elemental composition of sequences. Recently, the review of Pfister et al. (2022) in *mSystems* highlighted the relevance of elemental usage patterns in protein and nucleotide sequences. We briefly summarize refs. 69, 70, and 71 in that article below:

- Elser et al. (2011), itself a review article, cited several older studies that infer resource limitation from amino acid content of protein sequences coded in genomes.
- Lv et al. (2008) drew an association between resource availability and atomic content of proteins. The word "proteome" is prominent in that article, but their Methods indicate that they analyzed genomic protein sequences, not protein expression levels.
- Vecchio-Pagan et al. (2017) analyzed the elemental content of protein sequences from metagenomes for human body sites.

This brief overview suggests that there is a continuing interest in using genomically coded protein sequences to test hypotheses about environmental constraints. Our study follows that tradition, but takes a step forward by relating the environment to reference proteomes for communities instead of individual species.

We have revised the Introduction as follows (Line 83):

Genomic analyses have revealed intriguing patterns of elemental usage in protein sequences from different species (see Pfister et al., 2022 and references therein). However, observations for communities rather than species can be compared more directly with environmental measurements.

L158: least unstable protein -> least unstable reference proteome? it's clearly not just one protein, right?

Changed as suggested. (Line 671)

L156: "relative stability fields" is hard to understand unless you are familiar with the geochemistry language. I'd guess most readers are not.

Relative stability just means lower energy; how to calculate that energy is the difficult part. We have added the following text including a citation to the textbook by Anderson and Crerar (1993) for interested readers:

The model described here is defined in terms of thermodynamic components (see Anderson and Crerar, 1993), which are mathematical abstractions that theoretically relate the elemental composition of reference proteomes to chemical variables such as Eh and pH but do not necessarily reflect actual mechanisms of protein synthesis or evolution. (Line 183)

We wrote overall formation reactions for the reference proteomes from thermodynamic components, calculated the Gibbs energies of the reactions as a function of pH and Eh at 25 °C, then identified the reference proteome with the lowest Gibbs energy to plot relative stability fields (Fig. 1a; for further details, see Materials and Methods: Thermodynamic calculations). Because the

Gibbs energies are positive except at the most reducing conditions (near the stability limit of water), the reference proteomes are unstable with respect to the thermodynamic components across most of the diagram, and “relative stability” denotes the least unstable one compared to the others. (Line 208)

L213: why not include host-associated measurements if Eh is also reported? I don't follow. Regardless, there is enough analysis in this paper that this sentence can be cut.

We have not been able to find any metaproteomes – either free-living or host-associated – that have been reported with Eh data and publicly available protein (i.e., not only peptide) sequences. This sentence has been deleted. (Line 281)

L244: might be worth giving a number. how much more reduced are typical membrane proteins? I remember some discussion in Dick 2014 on this topic.

Added (Line 311):

For instance, in the yeast *Saccharomyces cerevisiae*, genomic sequences for proteins localized to the cytoplasm and plasma membrane have mean Z_C of -0.127 and -0.188 , respectively (Dick, 2014).

3. Statistical analysis -

L393-403 I found this paragraph somewhat repetitive with the previous.

This has been trimmed down to two nonrepetitive sentences that have been merged with the previous paragraph. (Lines 484-497)

3. Discussion.

- A major finding is that Archaea show uncertain, often negative Eh7-Z_C correlations. One explanation for this result is that the reference proteomes for Archaea are simply lower quality than those for bacteria. Many factors can limit the automated annotation of protein sequences from closed genomes, for example annotation of transcription start sites is ambiguous, as is identification of stop-codon suppression (e.g. for selenocysteine, pyrrolysine). See this reference for an example D. R. Gelsinger, et al., Nucleic Acids Res. 48, 5201-5216 (2020).

Thanks for this reference. We have added this sentence (Line 604):

In addition, reference proteomes for archaea are likely to be lower quality because automated annotation pipelines may not be current with recently discovered translation mechanisms (Gelsinger et al., 2020).

- A related concern to the above. Uniprot contains reference proteomes for ≈ 350 Archaea and ≈ 8800 bacteria. Perhaps the bacteria are better represented by (more closely related to?) their best-hit reference proteome than the Archaea analyzed on average?

This sounds right. We have restored and revised a sentence that was deleted in a previous revision (Line 601):

Across all datasets, the average percentages of genus-level classifications made by the RDP Classifier that were mapped to the NCBI taxonomy are 86% for bacteria and 77% for archaea, suggesting that archaeal communities are not as well represented by available reference proteomes.

L417: "from thermodynamics" -> "from our thermodynamic model of relative coding sequence stability" ... there is more than one reasonable thermodynamic model here.

This and the next sentence were repetitive with the first paragraph of the Discussion and have been deleted. (Lines 527-530)

L446-462: I am not sure I understand whether dispersal limitation is "good" or "bad" for your hypothesis. It took me two reads to understand that you think dispersal is "good" because it allows for the well-adapted microbes to get to the right redox niche, yes? But one could also imagine that lack of dispersal gives "time" for evolutionary sequence adaptation, or preferential growth of those microbes that are well-adapted. I think it is worth being a bit more explicit and clear here.

Considering the Reviewer's comment, and after further reflection, we recognize that our original interpretation doesn't hold up, or at least requires assumptions that are getting beyond the scope of our study. We had interpreted dispersal limitation as "bad" for our hypothesis, not only because organisms can't find their redox niche but also (more speculatively) because they don't face selective pressure through competition. Although getting stuck in one place gives organisms more time to adapt, they don't have to compete with other species moving through the environment. Under that interpretation, the lack of competition provides no drive for selection of well-adapted organisms. Ultimately, the "time" argument for adaptation may win out, and other authors have drawn associations between dispersal limitation and deterministic selection (Bottos et al., 2018).

We have removed the lines about dispersal limitation from the Discussion and from the Importance statement in the Abstract (Lines 562 and 46). We also removed the speculation about lower connectivity in sediments (Line 559).

More broadly, on the same L446-462... you convinced me that the point of this exercise is to ask whether you see a strong statistical association between coding sequence Z_C and Eh7. Figure 6 makes a good case from binomial statistics (for bacteria), but does that worldview permit such close inspection of individual datasets? Doesn't that amount to testing additional hypotheses? Your methods section claims there is only one hypothesis...

Multiple hypothesis testing frequently occurs in multivariate ecological studies. For example, this question poses nine hypotheses: which combinations of environmental factors a, b, and c and variables x, y, and z are correlated? In contrast, we proposed a hypothesis about the association between one factor (Eh7) and one variable (Z_C). The local correlations and global correlations are different tests (or "trials") of the same hypothesis. Binomial tests just indicate whether the collective outcome of individual trials at local and global scales is significant (assuming coin-flip probabilities).

Here we attempt to rephrase the Reviewer's question: can one hypothesis explain phenomena that occur at different length scales? It is possible in physics; the universal law of gravitation explains apples falling from trees as well as the orbits of planets.

"We have explored one hypothesis in this study" has been changed to "We subjected one hypothesis to multiple trials by assessing correlations between Eh7 and Z_C obtained from different local and

global datasets. Alternative hypotheses using Eh or O₂ concentration instead of Eh₇ were only tested for the pan-environmental global dataset.” (Line 851)

Added to Conclusion (Line 621):

These results suggest that chemical differences between protein sequences occurring at the millimeter scale or 1000-km scale may, to some extent, be explained by a single hypothesis of energy minimization shaping protein evolution.

L464-479: I feel you are getting ahead of yourself speculating about Archaeal physiology given the methodological concerns related to Archaeal ref. proteomes. A good opportunity to trim the text here.

Please note that the discussion about physiology was previously added in response to Reviewer #1, but we are responsible for any errors. The sentence about physiology has been deleted (Line 587).

4. Conclusion does a good job of tying the paper together.

L493-497: I got a little lost here. I don't understand how metaproteomics and 16S are related exactly.

This has been revised (Line 625):

Our conclusions are based on protein sequences from reference genomes and do not account for protein abundances. We could not find metaproteomic datasets that would permit comparing protein abundances with Eh measurements. Analysis of data from selected studies that reported both metaproteomes and 16S rRNA gene sequences revealed a positive but weak correlation between metaproteomic and 16S-based estimates of carbon oxidation state.

L499: Would rephrase here. It is worth being explicit that bacteria are typically ≈50% protein by dry weight, so elemental composition is strongly related to protein expression. You made an assumption that coding sequences are a good estimate for expressed proteins. This is obviously problematic but a nice starting point. And, moreover, it gives surprising results.

For a bulk cellular energy budget, evolution “cares” about the most abundant proteins, but there might be other reasons to care about the elemental composition of lowly expressed proteins. Bédard et al. (2022) stated that “highly expressed proteins evolve slower than lowly expressed ones.” This implies that when the environment changes, making adjustments to the elemental composition of lowly expressed proteins would be a faster way to reduce costs.

This is our revision. Please note that this paragraph has been moved from the Conclusions to the Discussion (Line 514):

A limitation of this study is that reference proteomes carry no information about protein abundance and cannot be used to make inferences about the effects of geochemistry on protein expression levels or whole-cell elemental composition. As proteins typically make up approximately half of cellular dry weight, the elemental composition of cells is strongly related to protein expression levels. On the one hand, highly expressed proteins require fewer total mutations than lowly expressed proteins for stoichiometric constraints to be accommodated by sequence evolution. On the other hand, lowly expressed proteins evolve faster than highly expressed proteins (Bédard et al., 2022). A tradeoff between abundance and evolvability suggests a reason for the sequences of both highly and lowly expressed proteins to be chemically adapted to environmental conditions and may

explain in part why community reference proteomes exhibit correlations with redox conditions despite having assumed equal contributions from proteins that actually vary widely in abundance.

L503-516: really nice closing. Honest and clear

Thank you.

Authors' Other Changes

- The title has been changed.
- Parts of the Abstract (most notably, the Importance section) have been revised.
- The Keywords have been updated.
- Parts of the Introduction have been revised for improved readability.
- An explanation for the gray area in Figure 1 has been added to the caption. (p 5, no line number)
- The value of slope mentioned in the “Theoretical background” section has been corrected (it was previously incorrectly taken from Fig. 2b instead of Fig. 1c). (Line 244)
- The last paragraph in the “Theoretical background” section has been revised. (Lines 248-266)
- The in-text citation of the authors Million and Raoult has been changed to the title and version of this resource: The “List of Prokaryotes According to Their Aerotolerant or Obligate Anaerobic Metabolism” (version 1.2). (Line 269)
- “amino acid analysis of bulk protein” has been added to the Conclusions as another means of testing the prediction. (Line 650)
- Background information about amplification bias has been moved from “Methodological limitations and justification” (Lines 882-885) to the Results section (Lines 451-454) and some repetition removed (Lines 885-887).
- Version numbers of R packages have been updated in the Data availability section. (Lines 905 and 909)
- Changes to word counts: Abstract 199 → 192, Importance 149 → 109, Main text 5665 → 5475, Materials and Methods 2738 → 2867.

Justification for Title Change

The previous title is unsatisfactory for two reasons. First, although “community reference proteomes” is a central concept for our study, its technical meaning may be unfamiliar to many readers and search engines. Second, “suggest” is used in an anthropomorphic sense. According to APA (2020), phrases such as “the results suggest” are acceptable. There is a large difference between these statements:

- Observations of bananas suggest adaptation to tropical conditions.
- Bananas suggest adaptation to tropical conditions.

Unfortunately, the previous title is more analogous to the second example above. Another weakness is the word choice of “geochemistry” instead of the specific kind of chemistry (i.e., redox potential, which is not limited to a geochemical context).

The titles for the previous revisions of the manuscript are listed below in reverse chronological order:

- Community reference proteomes suggest widespread influence of geochemistry on bacterial protein evolution [previous title]

- Reference proteomes for bacterial communities reveal chemical adaptation to redox conditions at a global scale [initial submission to *mSystems*]
- Local and global chemical shaping of bacterial communities by redox potential [preprint on bioRxiv]

The previous titles have in common an idea expressed by “influence”, “adaptation”, or “shaping”. Based in part on these considerations, we have retitled the manuscript “Community- and genome-based evidence for a shaping influence of redox potential on bacterial protein evolution”.

References added to manuscript

Anderson G. M. and Crerar D. A., (1993) *Thermodynamics in Geochemistry: The Equilibrium Model*. Oxford University Press, New York. <https://doi.org/10.1093/oso/9780195064643.001.0001>

Bédard C., Cisneros A. F., Jordan D. and Landry C. R. (2022) Correlation between protein abundance and sequence conservation: what do recent experiments say? *Current Opinion in Genetics & Development* **77**, 101984. <https://doi.org/10.1016/j.gde.2022.101984>

Pfister C. A., Light S. H., Bohannon B., Schmidt T., Martiny A., Hynson N. A., Devkota S., David L. and Whiteson K. (2022) Conceptual exchanges for understanding free-living and host-associated microbiomes. *mSystems* **7**, e01374-21. <https://doi.org/10.1128/msystems.01374-21>

References removed from manuscript

Dick J. M. (2021) Water as a reactant in the differential expression of proteins in cancer. *Computational and Systems Oncology* **1**, e1007. <https://doi.org/10.1002/cso2.1007>
[This reference has been replaced by a persistent URL for the canprot package on CRAN.]

Wasserstein R. L. and Lazar N. A. (2016) The ASA's statement on *p*-values: Context, process, and purpose. *The American Statistician* **70**, 129-133. <https://doi.org/10.1080/00031305.2016.1154108>
[This reference has recommendations for reporting statistical methods, which are not explicitly cited in most methods sections that we have seen.]

References cited in this response

APA (2020) *Publication manual of the American Psychological Association* (7th ed.). <https://doi.org/10.1037/0000165-000>

Bahram M., Hildebrand F., Forslund S. K., Anderson J. L., Soudzilovskaia N. A., Bodegom P. M., Bengtsson-Palme J., Anslan S., Coelho L. P., Harend H., Huerta-Cepas J., Medema M. H., Maltz M. R., Mundra S., Olsson P. A., Pent M., Pöhlme S., Sunagawa S., Ryberg M., Tedersoo L. and Bork P. (2018) Structure and function of the global topsoil microbiome. *Nature* **560**, 233-237. <https://doi.org/10.1038/s41586-018-0386-6>

Bottos E. M., Kennedy D. W., Romero E. B., Fansler S. J., Brown J. M., Bramer L. M., Chu R. K., Tfaily M. M., Jansson J. K. and Stegen J. C. (2018) Dispersal limitation and thermodynamic constraints govern spatial structure of permafrost microbial communities. *FEMS Microbiology Ecology* **94**, fiy110. <https://doi.org/10.1093/femsec/fiy110>

Bricker O. P. (1982) *Redox potential: Its measurement and importance in water systems*. In: Minear R. A. and Keith L. H. (Ed.), *Inorganic Species*, Academic Press. <https://doi.org/10.1016/B978-0-12-498301-4.50007-4>

Dick J. M. (2014) Average oxidation state of carbon in proteins. *Journal of the Royal Society, Interface* **11**, 20131095. <https://doi.org/10.1098/rsif.2013.1095>

Dick J. M. and Shock E. L. (2011) Calculation of the relative chemical stabilities of proteins as a function of temperature and redox chemistry in a hot spring. *PLOS One* **6**, e22782. <https://doi.org/10.1371/journal.pone.0022782>

Elser J. J., Acquisti C. and Kumar S. (2011) Stoichiogenomics: the evolutionary ecology of macromolecular elemental composition. *Trends in Ecology & Evolution* **26**, 38-44. <https://doi.org/10.1016/j.tree.2010.10.006>

Gelsinger D. R., Dallon E., Reddy R., Mohammad F., Buskirk A. R. and DiRuggiero J. (2020) Ribosome profiling in archaea reveals leaderless translation, novel translational initiation sites, and ribosome pausing at single codon resolution. *Nucleic Acids Research* **48**, 5201-5216. <https://doi.org/10.1093/nar/gkaa304>

Gittins D. A., Bhatnagar S. and Hubert C. R. J. (2023) Environmental selection and biogeography shape the microbiome of subsurface petroleum reservoirs. *mSystems* , e00884-22. <https://doi.org/10.1128/msystems.00884-22>

Lv J., Li N. and Niu D.-K. (2008) Association between the availability of environmental resources and the atomic composition of organismal proteomes: Evidence from *Prochlorococcus* strains living at different depths. *Biochemical and Biophysical Research Communications* **375**, 241-246. <https://doi.org/10.1016/j.bbrc.2008.08.011>

Masuda T., Saito N., Tomita M. and Ishihama Y. (2009) Unbiased quantitation of *Escherichia coli* membrane proteome using phase transfer surfactants. *Molecular & Cellular Proteomics* **8**, 2770-2777. <https://doi.org/10.1074/mcp.M900240-MCP200>

Vecchio-Pagan B., Bewick S., Mainali K., Karig D. K. and Fagan W. F. (2017) A stoichioproteomic analysis of samples from the Human Microbiome Project. *Frontiers in Microbiology* **8**, 1119. <https://doi.org/10.3389/fmicb.2017.01119>

February 28, 2023

Dr. Jeffrey M. Dick
Central South University
School of Geosciences and Info-Physics
Changsha
China

Re: mSystems00014-23R1 (Community- and genome-based evidence for a shaping influence of redox potential on bacterial protein evolution)

Dear Dr. Jeffrey M. Dick:

Thank you for your continued attention to the reviewers suggestions and documentation of changes to the manuscript in response. I am hereby now accepting this interesting work and will be very happy to see it published soon!

I am forwarding it to the ASM Journals Department. For your reference, ASM Journals' address is given below. Before it can be scheduled for publication, your manuscript will be checked by the mSystems production staff to make sure that all elements meet the technical requirements for publication. They will contact you if anything needs to be revised before copyediting and production can begin. Otherwise, you will be notified when your proofs are ready to be viewed.

If you would like to submit a potential Featured Image, please email a file and a short legend to msystems@asmusa.org. Please note that we can only consider images that (i) the authors created or own and (ii) have not been previously published. By submitting, you agree that the image can be used under the same terms as the published article. File requirements: square dimensions (4" x 4"), 300 dpi resolution, RGB colorspace, TIF file format.

We recognize that the video files can become quite large, and so to avoid quality loss ASM suggests sending the video file via <https://www.wetransfer.com/>. When you have a final version of the video and the still ready to share, please send it to mSystems staff at msystems@asmusa.org.

Thank you once again for submitting this interesting work to mSystems.

Sincerely,

Christopher Schadt
Editor, mSystems

Journals Department
E-mail: mSystems@asmusa.org